# Diverse neuronal activity patterns contribute to the control of distraction in the prefrontal and parietal cortex

**Panagiotis Sapountzis**[1,2]*, **Alexandra Antoniadou**[1,2]¤, **Georgia G. Gregoriou**[1,2]*

**1** Institute of Applied and Computational Mathematics, Foundation for Research and Technology Hellas, Heraklion, Crete, Greece, **2** Department of Basic Sciences, Faculty of Medicine, University of Crete, Heraklion, Crete, Greece

¤ Current Address: Centre de Recerca Matemàtica, Edifici C, Campus Bellaterra, Bellaterra, Spain
* pasapoyn@iacm.forth.gr (PS); gregoriou@uoc.gr (GGG)

**Data Availability Statement:** All data files that reproduce the main and supplementary figures are available from the zenodo database (accession

## Abstract

Goal-directed behavior requires the effective suppression of distractions to focus on the task at hand. Although experimental evidence suggests that brain areas in the prefrontal and parietal lobe contribute to the selection of task-relevant and the suppression of task-irrelevant stimuli, how conspicuous distractors are encoded and effectively ignored remains poorly understood. We recorded neuronal responses from 2 regions in the prefrontal and parietal cortex of macaques, the frontal eye field (FEF) and the lateral intraparietal (LIP) area, during a visual search task, in the presence and absence of a salient distractor. We found that in both areas, salient distractors are encoded by both response enhancement and suppression by distinct neuronal populations. In FEF, a larger proportion of units displayed suppression of responses to the salient distractor compared to LIP, with suppression effects in FEF being correlated with search time. Moreover, in FEF but not in LIP, the suppression for the salient distractor compared to non-salient distractors that shared the target color could not be accounted for by an enhancement of target features. These results reveal a distinct contribution of FEF in the suppression of salient distractors. Critically, we found that in both areas, the population level representations of the target and singleton locations were not orthogonal, suggesting a mechanism of interference from salient stimuli.

## Introduction

The ability to focus our attention on stimuli that are relevant to our current behavioral goals is largely dependent on our capacity to ignore irrelevant distractors. This task often becomes harder when distractors are more conspicuous by virtue of their physical properties, such as color popouts and bright or high-contrast objects. Imagine encountering a red car while searching for your black car among other black cars in a parking lot. Whether attention is captured automatically by such salient distractors is strongly debated [1,2]. Stimulus-driven theories propose that certain salient stimuli capture attention irrespective of the observer's intentions. This view is supported by psychophysical studies demonstrating that the presence

numbers https://zenodo.org/records/14577061, https://zenodo.org/records/14577123).

**Funding:** This research was supported by a grant from the Hellenic Foundation for Research and Innovation (HFRI-https://www.elidek.gr/en/homepage/) and the GSRT under the "1st Call for H.F.R.I. Research Projects to support Post-Doctoral Researchers" (Project No 1199) (to P.S.) and in part by the HFRI under the action "Basic Research Financing (Horizontal support for all Sciences), National Recovery and Resilience Plan (Greece 2.0)" (Project: 14941), EU NextGenerationEU (to G.G.G). The funders did not play any role in the study design, data collection and analysis, decision to publish, or preparation of the manuscript.

**Competing interests:** The authors have declared that no competing interests exist.

**Abbreviations:** AUROC, area under the ROC curve; ERP, event-related potential; FEF, frontal eye field; LIP, lateral intraparietal; MSE, mean square error; PEV, percentage of explained variance; RF, receptive field; ROC, receiver operating characteristic; SVM, support vector machine.

of color popouts slows visual search even when they are task irrelevant [3–5]. Conversely, goal-driven accounts suggest that attention is not automatically driven to a salient stimulus and that only stimuli matching the target features will draw attention [6,7]. The "signal suppression hypothesis" reconciles these opposing views by stating that although singleton distractors may produce a salience signal, they can be effectively suppressed through cognitive control [8,9]. In line with this, event-related potential studies in humans have provided evidence of salient distractor suppression manifested as the distractor positivity ($P_D$) component at posterior electrode sites [8,10]. However, it has been questioned whether $P_D$ can truly be considered an electrophysiological signature of active salience suppression [11,12]. Moreover, given that in displays with an oddball distractor the other distractors share features of the target, it has been argued that most singleton suppression effects can be rather explained by an enhancement of target features shared by the non-salient distractors [12,13]. Thus, the neuronal mechanisms of salient distractor encoding and/or suppression during visual search remain elusive.

Invasive recordings in monkeys that afford high spatiotemporal resolution have provided important insights at the neuronal level; however, results remain fragmented with different findings in different brain areas. Specifically, it has been shown that in the visual cortex, a pop-out color distractor induces a brief enhancement in activity followed by a pronounced suppression [14], whereas in higher order areas in the prefrontal and parietal cortex, only suppression effects have been reported [15,16]. These results suggest distinct signatures of salient distractor encoding in early and late stages of visual processing with activity in higher order areas reflecting and perhaps mediating only the effective suppression of salient irrelevant stimuli in line with behavioral demands. However, the latter studies reported average activity across trials and across the entire sample of recorded neurons that may result in loss of critical information. It is thus possible that in prefrontal and parietal areas, salient distractors are encoded similarly to visual cortex, albeit by distinct subpopulations with specific functional properties. Should this be the case, one could expect to find different groups of neurons in prefrontal (PFC) and/or parietal cortex; some encoding the salient stimulus through an early activity enhancement, other showing a later suppression of activity for the salient distractor and possibly yet other neurons displaying an early enhancement followed by a later suppression. Such a finding would indicate that at the neuronal level, the early encoding of the salient distractor is not unique to purely visual areas. It is therefore still unclear whether and how neuronal subpopulations in areas that have been proposed to hold an attentional priority map, such as the frontal eye field (FEF) [17] and the lateral intraparietal (LIP) area [18], encode salient irrelevant stimuli during search.

More importantly, if both target and salient distractor locations are encoded in the attentional priority map, the relationship between the 2 representations is yet to be examined. In recent years, there has been a paradigm shift that emphasizes representations of cognitive and other variables at the level of neuronal populations [19,20]. Within this framework, it is acknowledged that information is encoded in distributed patterns of population activity, even when it is not present in individual neurons. The geometry of these representations provides a description of how information is encoded. How the population level representations for targets and salient distractor locations relate to each other has not been addressed; yet, it is critical in order to understand how locations of potential interest are encoded in the brain and how interference from competing stimuli is avoided.

To address these questions, we performed simultaneous extracellular recordings from 2 key regions involved in the encoding of attentional priority, the FEF and LIP, during a free viewing visual search task with and without a singleton distractor. We find that both areas comprise distinct neuronal subpopulations that exhibit a mixture of temporal selectivity profiles, ranging from singleton suppression to singleton enhancement. Notably, we show that distractor

suppression is correlated with behavior in FEF, but not in LIP, with suppression being more pronounced in trials where animals are faster at locating the target. Critically, we find that in FEF, the response reduction for the singleton distractor is not due to an enhancement for non-singleton distractors that share features with the target. Finally, we highlight the role of mixed selective units in the encoding of the target and salient distractor location, and we show that although target and singleton representations are largely independent, a shared representation exists suggesting that the 2 representations are not orthogonal. Interestingly, these shared neural activity subspaces are composed of weighted combinations of units with diverse selectivity, rather than segregated subpopulations. Overall, our results provide novel insights into how behaviorally relevant and irrelevant, distracting stimuli are encoded, both at the local, as well as at the global population level.

## Results

We trained 2 monkeys in a free-viewing visual search task (Fig 1). At the beginning of each trial, animals were required to fixate on a central spot that was subsequently replaced by a cue stimulus corresponding to the search target. Following a variable delay period, an array of 8 stimuli appeared on the screen, one of which was the target. Animals were allowed to freely scan the array to locate the target and were rewarded after fixating it for 700 ms.

To examine the effect of a salient distractor on behavior and neuronal activity, we used 3 types of displays. In singleton present displays (Fig 1A), a singleton popped out by virtue of its color, i.e., a red, green, or yellow singleton among green, red, or blue stimuli, respectively. The location and identity of the singleton was randomized in each trial and was not known at the beginning of the trial. Critically, the singleton never served as the target throughout the training and recording sessions. Singleton present trials were randomly interleaved with singleton absent trials in which no singleton was present in the array and all stimuli shared the same color (Fig 1B), or mixed-color trials in which the salient and the 2 flanking stimuli of Fig 1A were kept constant and all other distractors shared no features with the target (Fig 1C).

Both animals effectively ignored the salient distractor as they seldom directed the first saccade to that stimulus (monkey PT in 1.5%, monkey FN in 1.4% of trials; percentage expected by random search 12.5%, $p < 0.001$, for both animals, chi-squared test) and the overall percentage of saccades to the salient distractors was low (monkey PT: 1.7%, monkey FN: 3.5%). Indeed, the animals' behavior indicated that the presence of the singleton effectively reduced the size of the search array. First, monkeys located the target faster in singleton present (Fig 1A) relative to singleton absent displays (Fig 1B). The median search time, calculated from the array onset to the beginning of target fixation, in singleton present displays was 330 ms for monkey PT and 541 ms for monkey FN, while in singleton absent displays it was 363 ms for monkey PT and 694 ms for monkey FN ($p < 0.001$, Wilcoxon rank sum test). In singleton present displays, the mean number of saccades executed to locate the target was 1.9 for monkey PT and 2.5 for monkey FN, whereas in singleton absent displays it was 2.1 for monkey PT and 2.8 for monkey FN ($p < 0.001$, Wilcoxon rank sum test). Moreover, monkeys tended to initiate the first saccade a few milliseconds earlier in singleton present compared to singleton absent displays (singleton present median: monkey PT: 181 ms, monkey FN: 242 ms; singleton absent median: monkey PT: 186 ms, monkey FN: 248 ms; $p < 0.001$, Wilcoxon rank sum test). These results confirm that the singleton stimulus was successfully ignored by the 2 animals.

### Singleton distractor suppression is evident in both FEF and LIP

We recorded neuronal activity from 534 units in FEF (301 from monkey PT and 233 from monkey FN) and 598 units in LIP (380 from monkey PT and 218 from monkey FN). We first

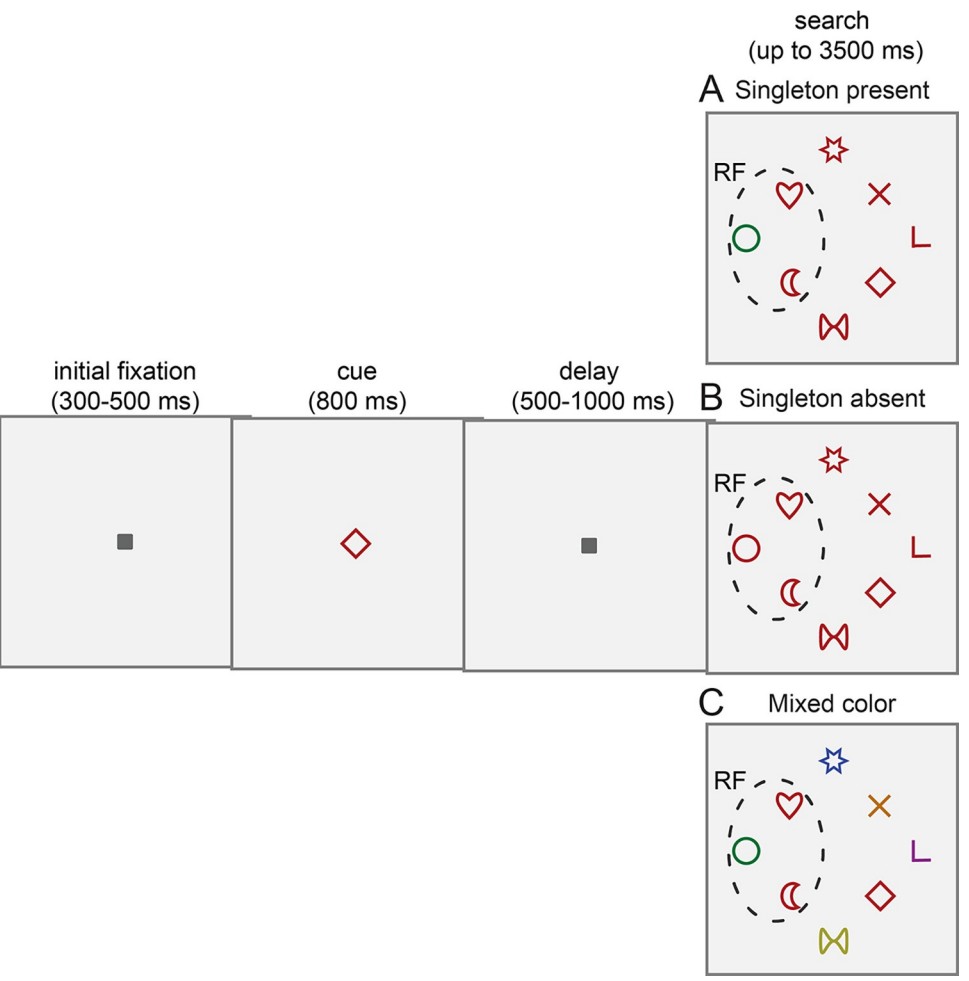

**Fig 1. Visual search task.** Animals were trained to fixate a central spot followed by a cue indicating the target stimulus. After a delay period, monkeys freely scanned the array to locate and fixate the target. Dashed lines indicate the hypothetical RF of a neuron. (A) In singleton present displays a stimulus popped out by virtue of its color, i.e., a red (green, yellow) distractor in an array of green (red, blue) stimuli. The location and identity of the salient distractor were randomized in each trial and were not known beforehand. Salient distractor trials were randomly interleaved with (B) singleton absent displays with no salient distractor (i.e., all distractors shared the target's color), and (C) with mixed-color displays in which the singleton and flanking stimuli of display A were kept the same and all other distractors shared no features with the target. RF, receptive field.

asked whether singleton distractors are more strongly suppressed relative to non-singleton ones in FEF and LIP. To this end, we considered singleton present displays (Fig 1A) and trials where a singleton or a non-singleton distractor fell inside the receptive field (RF) of the recorded units before the first saccade. To isolate the effects of stimulus encoding from those of saccade execution, we considered only trials in which the first saccade was made away from the RF. In FEF, average population responses to the salient distractor were significantly reduced relative to a non-salient distractor in the RF by 10.6% (paired $t$ test, $p < 0.001$, t = 5.2, df = 218, 150–200 ms following array onset; Fig 2A). These effects emerged before the onset of the first saccade (S1 Fig). We quantified the difference between responses in the 2 conditions at the level of individual units by calculating a modulation index (see Methods). The distribution of indices quantifying the response difference between the salient and a non-salient distractor in RF was significantly shifted toward negative values indicating a significant

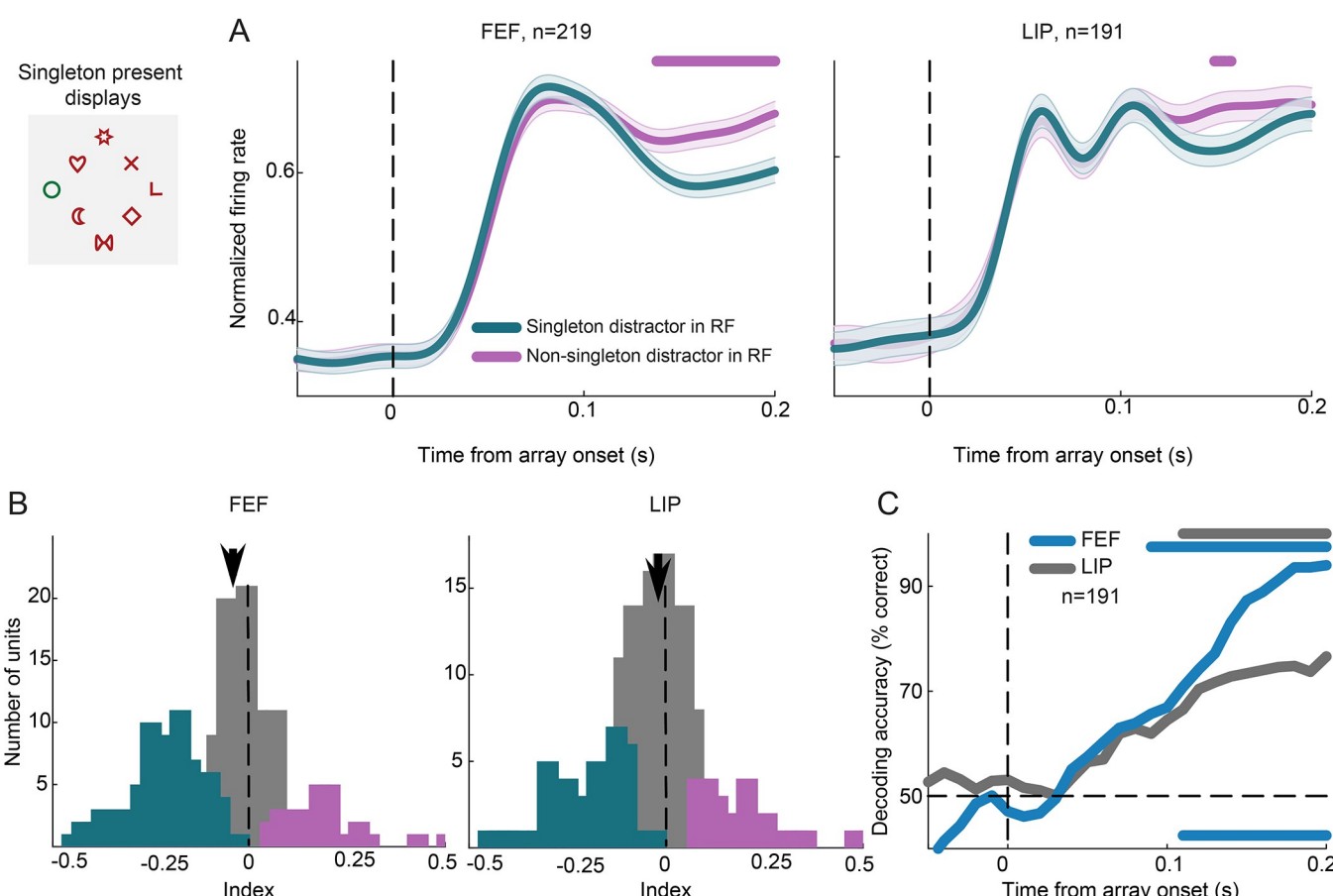

**Fig 2. Effect of singleton distractor on neuronal firing in FEF and LIP.** Responses in singleton and non-singleton distractor in RF were contrasted in singleton present displays (example display shown on top left; dashed lines show the hypothetical RF of a neuron). (A) Normalized population average firing rates aligned to array onset in singleton present displays, with the singleton (green) and a non-singleton (magenta) distractor in the RF, in FEF (left) and LIP (right). Error bars (shaded area around each line) represent ±SEM. The horizontal line at the top of each graph indicates periods with significant differences between the 2 conditions (permutation test, $p < 0.05$). (B) Distribution of modulation indices quantifying the difference between responses to the singleton and non-singleton distractors at the level of individual units in FEF (left) and LIP (right). Colored bars correspond to units with significant suppression (green) or enhancement (magenta) of responses for the singleton distractor (two-sample $t$ test, $p < 0.05$). Arrows indicate the median of each distribution (FEF: −0.04, LIP: −0.02). (C) Decoding accuracy over time for singleton or non-singleton distractor in the RF, in FEF (blue) and LIP (gray). The horizontal dashed line indicates the chance level (50%). Horizontal colored lines at the top of the graph indicate periods of significant decoding accuracy compared to chance for each area (cluster-based permutation test, $\alpha = 0.05$). The horizontal line at the bottom shows time periods with significant differences between the 2 areas (cluster-based permutation test, $\alpha = 0.05$). Source data are available at https://zenodo.org/records/14577061. FEF, frontal eye field; LIP, lateral intraparietal; RF, receptive field.

suppression in activity for the salient distractor in RF (Fig 2B; Wilcoxon signed-rank test, $p < 0.001$, z = −4.8, effect size r = −0.33; see Methods on how effect sizes were calculated in the non-parametric case).

In LIP, responses to the salient relative to a non-salient distractor were suppressed by 4.3% (paired $t$ test, $p = 0.04$, t = 2.1, df = 190, 150–200 ms following array onset; Fig 2A). The distribution of indices quantifying the difference between the salient and non-salient distractor was significantly shifted toward negative values (Wilcoxon signed-rank test, $p < 0.05$, z = −2.2, effect size r = −0.16). Moreover, approximately a fourth of LIP units exhibited significant distractor suppression (48 out of 191, two-sample $t$ test at $\alpha = 0.05$; Fig 2B), whereas 37% (81/219) of units showed significant suppression in the FEF, indicating a higher proportion of units with significant singleton suppression in the FEF ($p < 0.01$, chi-squared test). By contrast, the

percentage of units with significant enhancement in the 2 areas was not significantly different (LIP: 28/191, 14.7%; FEF 30/219, 13.7%; $p$ = 0.8, chi-squared test). Comparison of the distribution of modulation indices in FEF and LIP showed that the suppression was marginally larger in FEF compared to LIP (Wilcoxon rank-sum test, $p$ = 0.049, z = 1.96, effect size r = 0.13). Data for each animal separately are shown in S2 Fig.

It should be noted that in case of sparse representations of information, results from averaging across trials and neurons can be misleading. Several studies have overcome this limitation by employing multivariate classification methods that consider the pattern of activity across populations of neurons [21–23]. To this end, we considered responses in the 2 conditions across all recorded neurons in each area and we applied a linear support vector machine (SVM) classifier to predict on a trial-by-trial basis whether the salient or a non-salient distractor was in the RF. As in the univariate analysis, we considered trials in which the first saccade was made away from the RF. In FEF, discrimination between salient and non-salient distractors was robustly significant (Fig 2C, blue line; cluster-based permutation test, α = 0.05), consistent with the univariate analysis results. In LIP, accuracies in classification between salient and non-salient distractors also reached significance (Fig 2C, gray line; cluster-based permutation test, α = 0.05), although performance was significantly lower relative to FEF (Fig 2C, horizontal blue line at the bottom of the graph; cluster-based permutation test, α = 0.05).

Overall, we found a larger proportion of units with significant suppression in the FEF and higher decoding accuracies compared to LIP. However, our results show that neurons whose activity for the singleton distractor is significantly modulated exist in both areas, suggesting that specific neuronal subpopulations may carry the relevant information.

## Salient distractor suppression or target feature enhancement?

Our results suggest that responses to the salient distractor are reduced relative to non-salient distractors, which is consistent with the signal suppression hypothesis. However, non-salient distractors shared the same color with the target. Therefore, an alternative interpretation could be that the difference in activity between salient and non-salient distractors is due to a global enhancement of target features for the latter rather than active suppression of the former, as previously suggested [12,13]. Indeed, we and others have shown that neuronal responses in both FEF and LIP are enhanced for stimuli that share the target's features during visual search [24–26]. Thus, it is possible that the observed reduction in response to a salient relative to a non-salient distractor does not reflect an active suppression mechanism, but it is rather a consequence of a global enhancement of the target's color shared by the non-salient distractors.

Although the 2 interpretations are not mutually exclusive, we tested whether the difference in responses between singleton present and singleton absent displays could be attributed to a global enhancement of target features only. To this end, we contrasted responses in salient distractor displays (Fig 1A) to those in mixed-color displays (Fig 1C). Critically, we sought to compare responses to a salient distractor within the RF to those to the exact same stimuli in the RF presented in a non-salient context (Fig 3, top). We reasoned that if the response difference between singleton present and singleton absent displays (Fig 2) had been exclusively due to an enhancement of target color for the non-salient distractors, no difference should be found in activity when comparing singleton present to mixed color displays given that the stimuli in the RF are identical. By contrast, if the salient distractor is actively suppressed in the singleton present displays, responses to a salient distractor in RF should be lower in singleton displays compared to those in mixed-color displays with identical RF stimuli. Note that neurons were included in the analysis if they fulfilled the minimum trials criterion in the 2 conditions (see Methods); thus, the populations in Figs 2 and 3 are not identical. In FEF, population

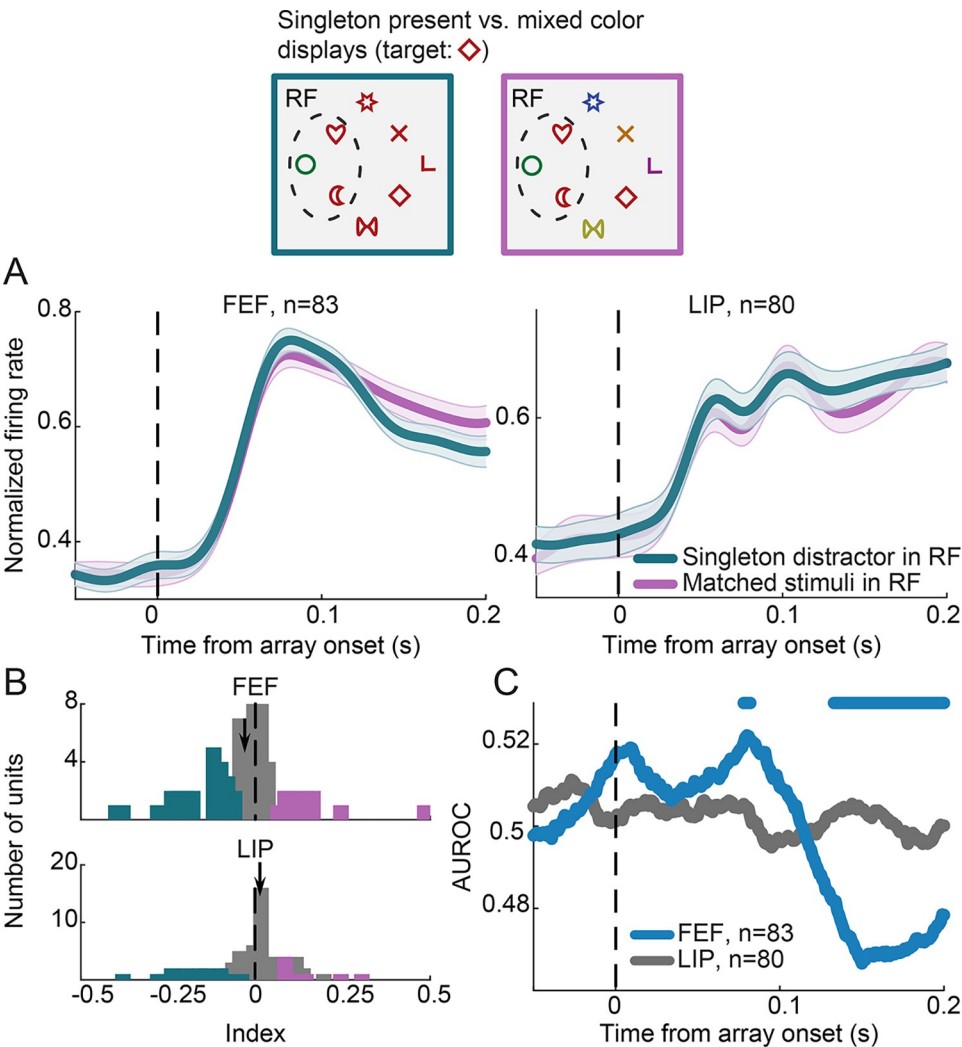

**Fig 3. Effect of singleton distractor relative to the exact same stimuli presented in a non-salient context.** Neuronal responses are compared between conditions such as those in the example displays shown on top. (A) Normalized average firing rates for the singleton distractor in the RF (green) relative to the exact same stimuli in a non-salient context (magenta) in FEF (left) and LIP (right). All other conventions as in Fig 2A. (B) Modulation indices quantifying the firing rate difference between the 2 conditions at the level of individual units in FEF (upper panel) and LIP (lower panel). Arrows indicate the median of each distribution (FEF: −0.03, LIP: −0.005). (C) The AUROC contrasting the 2 conditions in FEF (blue) and LIP (gray). Horizontal blue line at the top shows periods with a significant difference from chance for the FEF indicating a significant modulation by the singleton distractor (permutation test corrected for false discovery rate, $p < 0.05$). Source data are available at https://zenodo.org/records/14577061. AUROC, area under the ROC curve; FEF, frontal eye field; LIP, lateral intraparietal; RF, receptive field.

average responses to a singleton distractor in the RF were suppressed relative to the same stimuli presented in a non-salient setting by 8.4% (150–200 ms following array onset; $p < 0.01$, paired $t$ test; Fig 3A, left). In addition, the distribution of modulation indices that quantified the difference between the 2 conditions at the level of individual units was shifted significantly toward negative values indicating lower responses on average for the salient distractor in RF in singleton displays (Fig 3B, top panel; $p = 0.03$, Wilcoxon signed-rank test). Moreover, the average across FEF units area under the ROC curve (AUROC) values contrasting the 2 conditions, were significantly lower than 0.5, indicating a significant reduction in response to the salient relative to the same but non-salient stimuli (Fig 3C, blue line; $p < 0.05$, permutation test

corrected for false discovery rate). By contrast, no significant difference between the salient and the non-salient distractor was found in LIP (Fig 3A, right). The distribution of modulation indices was not significantly different from zero (Fig 3B, bottom panel; $p = 0.48$, Wilcoxon signed-rank test) and AUROC values were not significantly different from 0.5 (Fig 3C, gray line; $p > 0.05$, permutation test corrected for false discovery rate).

These results suggest that in FEF, but not in LIP, singleton distractors are suppressed beyond the level of suppression of irrelevant distractors presented in a non-salient setting. This is in agreement with the signal suppression hypothesis. In LIP, on the other hand, the modest reduction in activity for salient distractors relative to non-salient ones may mainly result from an enhanced activity for stimuli sharing the target color.

## Distinct neuronal subpopulations manifest unique selectivity profiles

We have previously shown that LIP comprises a heterogeneous population of neurons with different contributions in encoding attentional priority [25]. Moreover, our current analysis on firing rate modulations for target and salient distractor stimuli suggests that individual neurons show distinct profiles of modulation (Fig 2B). Motivated by these findings, we wished to investigate the potential contribution of specific neuronal subpopulations in the encoding of salient distractors. To this end, we compared responses between the singleton present and singleton absent displays (Fig 4, top left). We identified visually responsive units (see Methods) and for each of those we calculated a t-value quantifying its selectivity for a salient versus a non-salient distractor within the RF. Units were included in the analysis if they exhibited significant t-values for at least 10 consecutive 1 ms bins. Similar to the analyses above, we considered only fixations before saccades away from the RF. Subsequently, we employed an unsupervised algorithm (PhenoGraph) to cluster units according to their t-value selectivity profiles [27]. The algorithm identified 3 clusters in FEF and 3 clusters in LIP, with unique temporal selectivity profiles (Fig 4A). In the first cluster, negative t-values (shown in blue) were more prominent indicating suppression in activity for the salient distractor (bottom of each panel in Fig 4A, cyan cluster). In the second cluster, positive t-values (shown in red) indicating an enhancement in activity for the salient distractor were evident in the initial response only, followed by mostly negative t-values (middle of each panel in Fig 4A, pink cluster). Finally, in the third cluster positive t-values were more prominent indicating a response enhancement for the salient distractor (top of each panel in Fig 4A, green cluster). Average firing rates calculated for each cluster further illustrate their distinct selectivity profiles (Fig 4B). The same distinct selectivity profiles were found in each animal separately (S3 Fig).

To investigate the time course of salient distractor enhancement and suppression, we subtracted the response to the salient distractor from that to the non-salient distractor, as described in [14]. We then estimated the latency of these effects as the first significant 5 ms bin ($p < 0.05$, t test with Bonferroni correction for multiple comparisons) in a series of 5 consecutive windows. The first subpopulation (cluster 1; cyan frame) exhibited a reduced response to the salient relative to a non-salient distractor. This reduction occurred significantly earlier in FEF relative to LIP (LIP: 95 ms, FEF: 60 ms, $p < 0.01$, permutation test). The second subpopulation (cluster 2; pink frame) demonstrated an early enhancement for the singleton, followed by a subsequent suppression. Singleton enhancement occurred at similar times in both areas (25 ms, $p = 0.25$, permutation test). By contrast, the subsequent singleton suppression in the activity of the same neuronal subpopulation emerged significantly earlier in FEF compared to LIP (LIP: 150 ms, FEF: 125 ms, $p < 0.01$, permutation test). Finally, the third subpopulation (cluster 3; green frame) exhibited a late enhancement for the salient relative to a non-salient

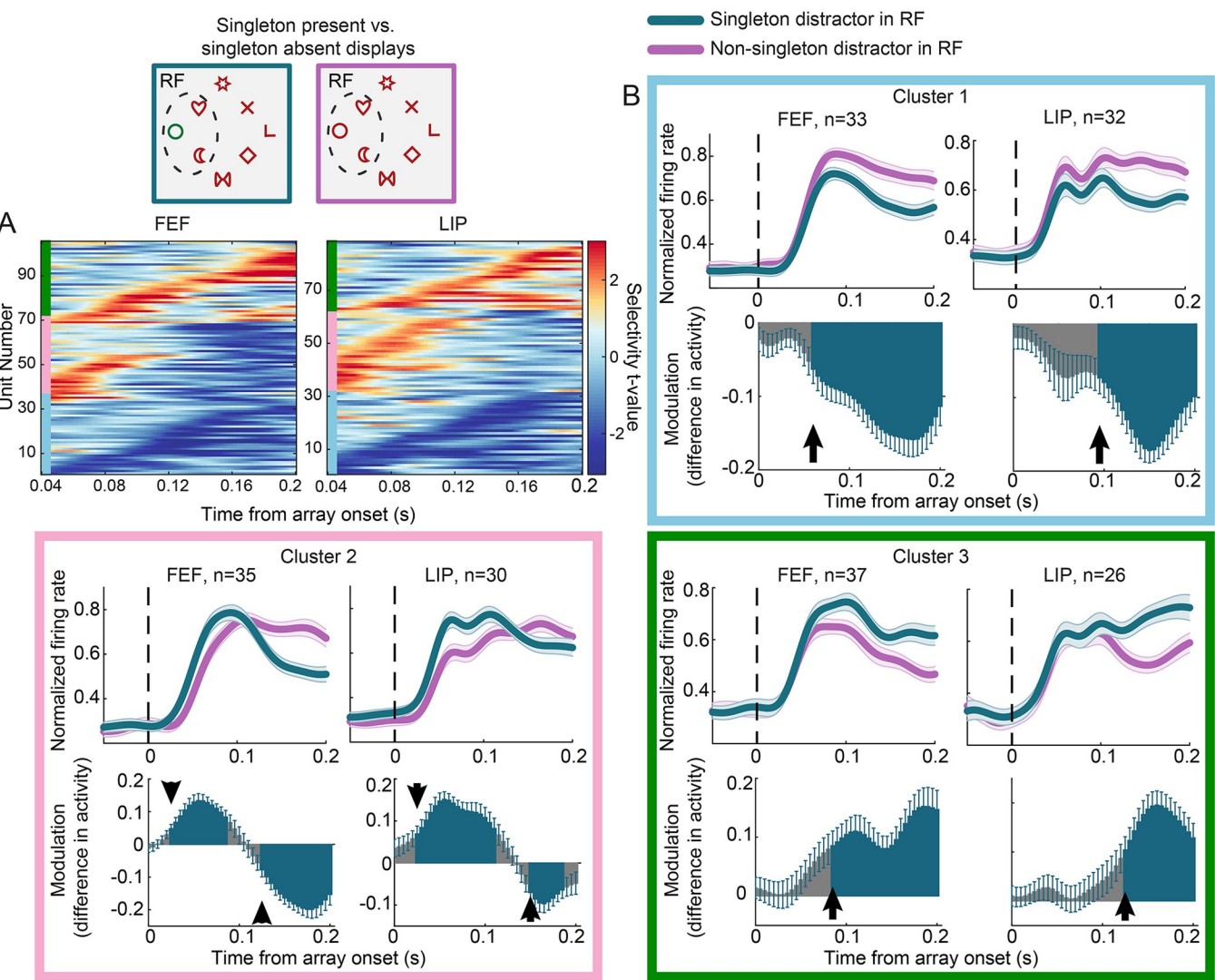

**Fig 4. Distinct subpopulations with unique selectivity profiles.** Responses in singleton present and singleton absent displays (example displays shown on top left) were contrasted. (A) Selectivity t-values across time for singleton vs. non-singleton distractors, clustered using the PhenoGraph algorithm in FEF (left) and LIP (right). Clusters are indicated by the color bars along the y-axis. Each row illustrates selectivity t-values of one unit sorted, within each cluster, by the time they first became significant. (B) The 3 clusters identified by the PhenoGraph algorithm in each region. Box colors correspond to those shown along the y-axis in panel A. Top panels show average normalized firing rates in singleton (green) and non-singleton (magenta) in the RF trials. To facilitate comparison, the average baseline activity was subtracted from each condition. All other conventions are as in Fig 2A. Bottom row panels show the difference in activity elicited by the singleton and non-singleton distractor in the RF over time. Error bars represent ±SEM. Arrows indicate the latency of significant singleton enhancement or suppression, defined as the first of 5 consecutive 5 ms significant bins. Significance in each time bin was determined by means of a one-sample *t* test with Bonferroni correction ($p < 0.05$). Source data are available at https://zenodo.org/records/14577061. FEF, frontal eye field; LIP, lateral intraparietal; RF, receptive field.

distractor. This difference in the latencies between the 2 areas was not significant (LIP: 125 ms, FEF: 85 ms, $p = 0.3$, permutation test).

Our findings demonstrate that different neuronal subpopulations manifest unique selectivity profiles that are not discernible in the average response of the entire population. These temporal selectivity profiles range from a late suppression of salient distractors (cluster 1) to an initial enhancement followed by a suppression (cluster 2), and an enhancement of salient distractors (cluster 3). In the case of singleton suppression, effects emerged earlier in the FEF,

consistent with its role in attentional selection. Similar selectivity profiles were found when analyzing responses from salient distractor present displays only (S4 Fig).

## Reduced singleton responses in FEF but not LIP correlate with behavior

One could argue that the observed activity suppression for salient distractors is due to processes that are irrelevant to behavior. We thus asked whether the level of activity for distractors was correlated to behavioral performance. To this end, we calculated the time it took animals to locate the target (from the onset of the array to the initiation of target fixation), and we applied a median split approach classifying trials as either fast or slow. In the analysis, we considered salient distractor displays (Fig 5, upper left) and only included saccades away from the RF. In FEF, responses to both salient and non-salient distractors were significantly reduced in fast relative to slow trials (Fig 5A; $p < 0.05$, permutation test; 150–200 ms following array onset) indicating that the more effective the suppression of a distractor, the faster the target is located. Moreover, at the level of individual FEF units, modulation indices contrasting responses between fast and slow trials were significantly shifted toward negative values for both salient and non-salient distractors, suggesting again reduced activity in response to

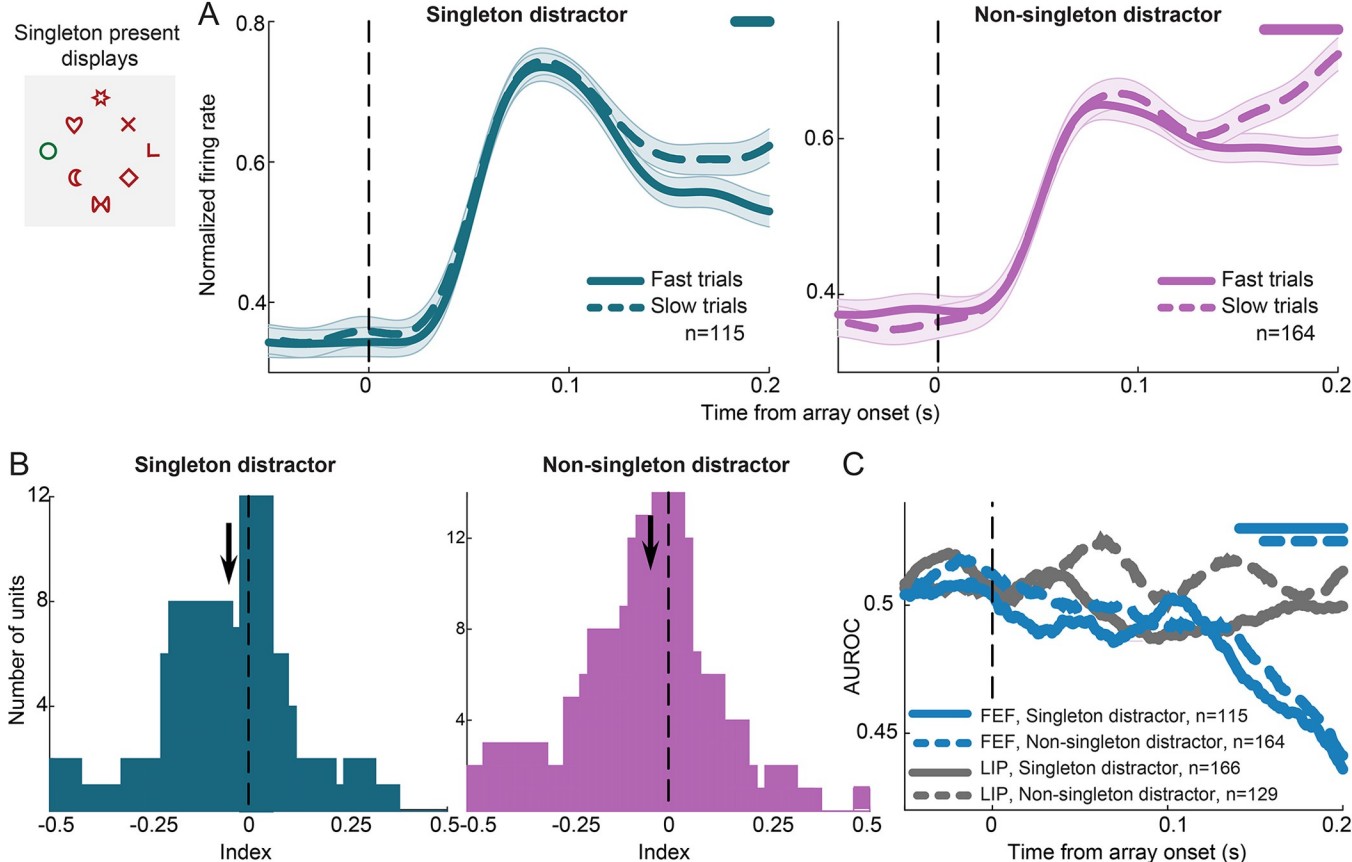

**Fig 5. Distractor suppression in FEF correlates with behavior.** Responses in fast and slow search trials were contrasted in singleton present displays (example display shown on top left). (A) FEF responses to singleton (left) and non-singleton (right) distractors were significantly reduced in fast (solid line) relative to slow/prolonged (dashed line) search trials, as indicated by the line at the top of each graph (permutation test, $p < 0.05$). (B) Modulation indices quantifying the difference between fast and slow trials at the level of individual FEF units for singleton (left) and non-singleton (right) distractors in the RF. Arrows indicate the median of each distribution (singleton distractor: −0.049, non-singleton distractor: −0.044). (C) The AUROC contrasting fast and slow trials in FEF (blue) and LIP (gray), for singleton (solid lines) and non-singleton (dashed lines) distractors in the RF. All other conventions are as in Fig 3C. Source data are available at https://zenodo.org/records/14577061. AUROC, area under the ROC curve; FEF, frontal eye field; LIP, lateral intraparietal; RF, receptive field.

distractors during fast trials (Fig 5B; Wilcoxon signed-rank test; salient distractor $p < 0.001$, z = −3.3, effect size r = −2.4; non-salient distractor $p < 0.001$, z = −4.1, effect size = −2.9). By contrast, no significant difference in activity was found in LIP between fast and slow trials when a distractor was in the RF, neither at the population level nor at the individual unit level (S5 Fig; Wilcoxon signed-rank test; salient distractor $p > 0.1$, z = 0.4, effect size r = 0.3; non-salient distractor $p > 0.1$, z = 0.7, effect size r = 0.5). These results were further quantified using an ROC analysis implemented to discriminate between responses to a distractor in the RF in fast and slow trials, for each unit. The average area under the ROC curve values in FEF were significantly lower than 0.5, indicating reduced responses to salient and non-salient distractors on fast trials (Fig 5C; solid and dashed blue lines, respectively; $p < 0.001$, permutation test corrected for false discovery rate). By contrast, the area under the ROC curve values in LIP did not significantly deviate from 0.5 for either the salient or the non-salient distractor (Fig 5C; solid and dashed gray lines, respectively).

Overall, our findings indicate a correlation between distractor suppression in the FEF and the time needed to locate the target. Therefore, suppression of salient stimuli implemented in the FEF seems to facilitate efficient target selection.

## Mixed-selectivity drives decoding at the population level

We have previously shown that the behavioral relevance of the RF stimulus (target or distractor) can be decoded from the FEF and LIP population activity albeit more robustly from FEF in line with the role of these areas in holding a map of attentional priority [25]. However, a map of attentional priority also encodes locations of salient stimuli. Indeed, we show that salient stimuli although irrelevant, are encoded by FEF and LIP subpopulations with different activity profiles. The question that arises is how the population representation of target and salient distractor locations emerges from neurons with distinct activity profiles. To answer this question, we first sought to examine whether the location of the target or salient distractor can be decoded from the pattern of population activity in FEF and LIP. Decoding accuracies were calculated in salient distractor displays (Fig 6, left) without considering the neurons' RF or the

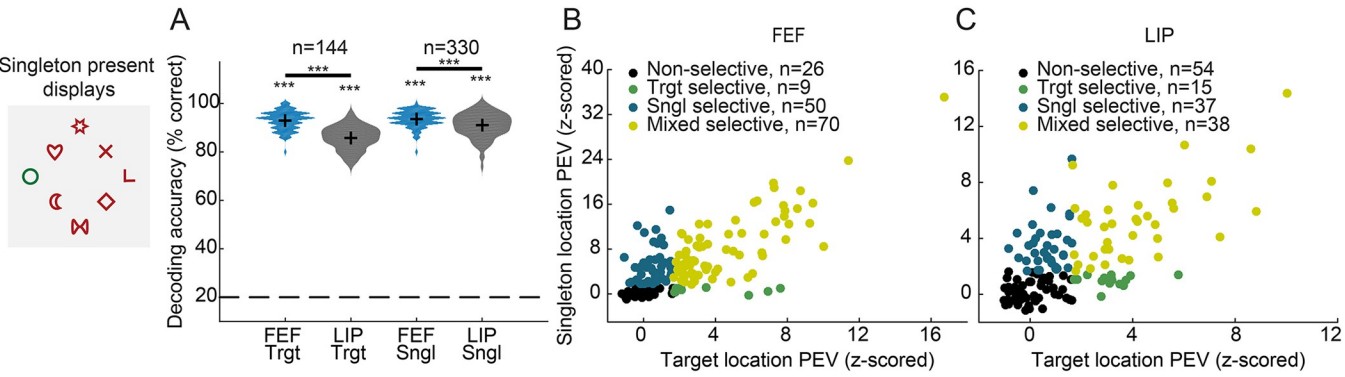

**Fig 6. Encoding of target and singleton distractor locations in FEF and LIP.** (A) Decoding accuracy of target and singleton locations calculated in the 150–200 ms interval after array onset in singleton present displays (example shown on the left). Violin plots show the distribution of decoding accuracies calculated over 50 resamples across different trials. The dashed horizontal line indicates chance accuracy. Decoding was carried out across the 3 left hemifield and 2 vertical meridian locations, thus chance was at 20%. Crosses indicate the mean of each distribution. Stars above each distribution show significant accuracy relative to chance (permutation test, $p < 0.001$). Stars over horizontal bars indicate difference between areas (permutation test, $p < 0.001$). To allow for comparison between areas, the number of units was stratified prior to accuracy calculation (50 resamples). (B) Selectivity of FEF units to target and singleton location as quantified by the PEV metric, calculated in the 150–200 ms interval after array onset. Units were characterized as target selective (green), singleton selective (blue), mixed selective (yellow), or non-selective (black). Significant selectivity was determined as z-scored PEV > 1.645 (one-tailed test). (C) Same as in B, but for LIP. Source data are available at https://zenodo.org/records/14577061. FEF, frontal eye field; LIP, lateral intraparietal; PEV, percentage of explained variance.

direction of the upcoming saccade. As we were interested in locations contralateral to the recording hemisphere, we considered the 5 left hemifield locations (including the 2 vertical positions), thus chance accuracy was at 20%. Interestingly, despite the more pronounced singleton suppression effects that were observed at the level of average activity in FEF, both target and singleton location were reliably decoded from both areas in the 150–200 ms interval after array onset with over 80% accuracy, although accuracies were significantly higher for the FEF (permutation test, $p < 0.001$, Fig 6A).

To assess how neuronal populations with distinct spatial selectivity profiles for target and singleton distractor contribute to the population code for target and singleton position, we employed the percentage of explained variance (PEV) metric. We quantified how much of the variance in the firing rate of each unit can be explained by the target's or the singleton distractor's location. For each unit, we assessed whether the PEV reached significance in the interval 150–200 ms after the array onset (z-scored PEV $> 1.645$, one-tailed test) and characterized units as (i) target selective; (ii) singleton selective; (iii) mixed selective (i.e., selective for both target and singleton distractor location); or (iv) non-selective. In FEF, 45.2% of units were classified as mixed selective, 32.3% as singleton selective, 5.8% as target selective and 16.8% as non-selective (Fig 6B). In LIP, 26.4% of units were characterized as mixed selective, 25.7% as singleton selective, 10.4% as target selective, and 37.5% as non-selective (Fig 6C). These results offer several insights. First, a larger proportion of mixed-selective units was sampled in FEF relative to LIP (45.2% versus 26.4%; $p < 0.001$, chi-squared test). Second, mixed-selective units had on average higher PEV values than units selective for target or singleton distractor location alone ($p < 0.001$, 2-way ANOVA with Tukey–Kramer correction for multiple comparisons, except for the comparison with target selective units in FEF, $p > 0.8$). Furthermore, these neurons drove to a large extent decoding accuracy at the population level (S6 Fig). Finally, fewer non-selective units were sampled in FEF relative to LIP (16.8% versus 37.5%; $p < 0.001$, chi-squared test). In summary, mixed-selective units appear to play a major role in the encoding of target and salient distractor location as they exhibit overall higher selectivity and drive decoding at the population level.

## Representations of the target and salient distractor locations do not occupy orthogonal subspaces

It is conceivable that a target and a salient distractor impose different behavioral requirements as the location of the former is encoded for selection while the location of the latter is encoded for avoidance. The question that emerges then is whether the representations of the relevant locations are overlapping or orthogonal in neuronal activity space as previously reported for different memory items [28]. If the location of the target and salient distractor are represented in independent subspaces, it may offer a coding system in which competing items are represented independently to avoid interference.

To test this hypothesis, we applied manifold optimization to jointly identify mutually orthogonal subspaces for the target and salient distractor locations. If the target and singleton distractor subspaces are indeed orthogonal, the cross-projected variance between the 2 should be close to zero. Conversely, a substantial cross-projected variance would suggest a lack of orthogonality. Subspaces were calculated in the 150–200 ms interval following array onset. We present results for five-dimensional subspaces. The variance explained in a certain projection was calculated as the variance captured in the five-dimensional subspace divided by the variance captured by the top 5 PCs (i.e., the maximum possible variance that can be captured by 5 dimensions; see Methods). The optimization procedure identified 2 orthogonal subspaces, one for the target (Orth-Target) and one for the salient distractor (Orth-Singleton), capturing

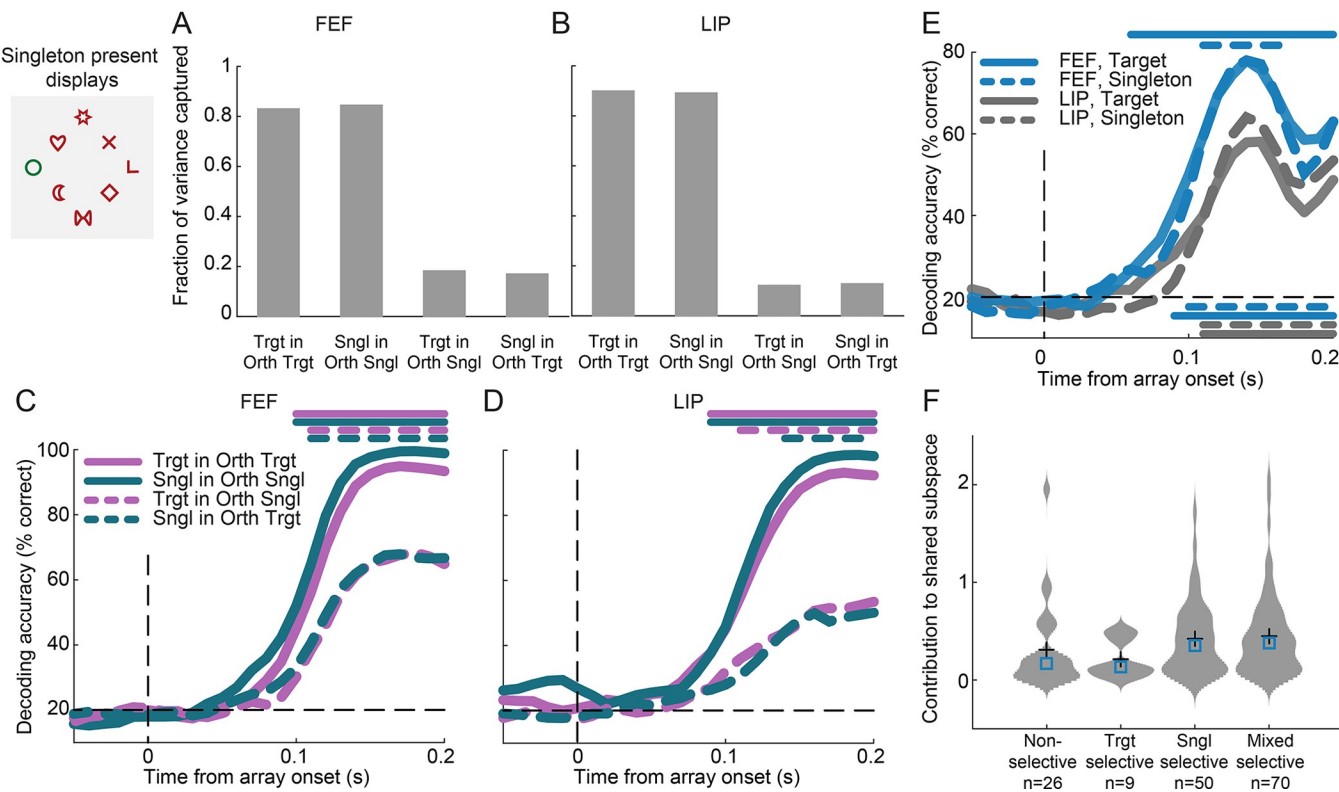

**Fig 7. Relationship of target and singleton distractor representations based on neural subspace analysis.** Responses were calculated in singleton present displays (example shown on top left). (A) Variance captured by projecting target and singleton FEF activity on the Orth-Target and Orth-Singleton subspaces. (B) Same for LIP. (C) Decoding accuracy for target/singleton distractor location using projected target (magenta) and singleton (green) FEF activity on same context (solid lines) and different context subspaces (dashed lines). The horizontal dashed line indicates chance level (20%). Lines at the top of the graph indicate periods of significant decoding accuracy relative to chance for each line (cluster-based permutation test, α = 0.05). (D) Same for LIP. (E) Decoding accuracy for target/singleton location using target (solid lines) and singleton (dashed lines) activity projected onto the shared subspace in FEF (blue) and LIP (gray). Horizontal lines at the top show time periods with significant differences between the 2 areas (cluster-based permutation test, α = 0.05). Lines at the bottom of the graph indicate periods of significant decoding accuracy relative to chance for each line (cluster-based permutation test, α = 0.05). (F) Contribution to the shared subspace of FEF units with different selectivities. Violin plots show the distribution of weights contributing to the shared subspace separately for each category. Crosses represent the mean of each distribution and boxes the median. Source data are available at https://zenodo.org/records/14577061. FEF, frontal eye field; LIP, lateral intraparietal.

more than 80% of the variance related to the location of target and salient distractor, respectively (FEF: Fig 7A, Orth-Target 83%, Orth-Singleton 84%; LIP: Fig 7B, Orth-Target 90%, Orth-Singleton 89%). Interestingly, however, we found substantial cross-projected variance. In FEF, the variance of target location related activity projected on Orth-Singleton was 18% (12% in LIP), and the variance of singleton location related activity projected on Orth-Target was 17% (13% in LIP). To test whether the cross-projected variance was task-relevant or simply reflected noise, we decoded the target and singleton location from the activity projected onto the subspaces. Decoding of projected activity on same context subspaces (target on Orth-Target or salient on Orth-Singleton) reached accuracies higher than 90% (Fig 7C and 7D, solid lines). More importantly, decoding of cross-projected activity was significantly above chance (Fig 7C and 7D, dashed lines), suggesting a lack of orthogonality. Notably, cross-projected variance and decoding performance were lower in LIP than in FEF, suggesting more independent representations in LIP.

Since target and singleton representations are not orthogonal, they must occupy a shared subspace. To identify such a subspace, we first applied manifold optimization to identify 2

exclusive subspaces that maximized the variance for the target- (singleton-) location while capturing less than 1% of the variance for singleton- (target-) location, respectively. Subsequently, we calculated the shared subspace by forcing it to be orthogonal to these exclusive subspaces, while jointly maximizing the target and singleton distractor variance. In FEF, the shared subspace captured 26% (23% in LIP) of projected target variance and 27% (19% in LIP) of projected singleton variance. Activity projected to the shared subspace was task relevant as decoding accuracy was high in both FEF and LIP (Fig 7E). However, the difference between areas was statistically significant (Fig 7E, lines at the top of the graph; cluster-based permutation test, α = 0.05), again indicating that target and singleton representations are more independent in LIP than in FEF.

The lack of orthogonality might not come as a surprise, given the large proportion of mixed-selective units found in the 2 areas (Fig 6C and 6D). Therefore, we asked whether conjunctive units contributed larger weights to the shared subspace (see Methods). We found that this was not the case. FEF units with different selectivity profiles did not exhibit distinct contributions to the shared subspace (Fig 7F; one-way ANOVA, $p$ = 0.2, F = 1.51, df = 154). The same was true for the shared subspace in LIP (S7 Fig). This finding indicates that the shared subspace is composed of a weighted combination of neurons, including those with nonsignificant selectivity, rather than a distinct subpopulation. This aligns with several recent studies suggesting that it is the pattern of activity across the population, rather than a segregated subpopulation, that identifies subspaces [29,30].

## Discussion

In this study, we performed simultaneous recordings from monkey FEF and LIP to discern their role in the encoding and modulation of salient, behaviorally irrelevant distractors. We employed a free viewing visual search task in which the identity of the target was known at the beginning of each trial and the singleton distractor never served as a target through training and recordings. Thus, attentional capture by the singleton distractor could in principle be prevented. Contrary to previous studies, which have shown only suppression of salient irrelevant stimuli in the prefrontal and parietal cortices, here, we report both salient distractor activity enhancement and suppression in the 2 areas. Moreover, we provide evidence of distinct contributions of the 2 areas, with FEF exhibiting a larger proportion of units with suppression to the salient, but irrelevant, distractor compared to LIP. This is consistent with the prominent role of the prefrontal cortex in suppressing conspicuously distracting stimuli [15,31–33] and its well-documented role in target selection [15,17,25,26]. Notably, distractor suppression was correlated with behavior in FEF, but not in LIP, as it was more pronounced in trials where animals were faster at identifying the target. This finding suggests that the ability to suppress irrelevant stimuli facilitates search behavior and is reflected in FEF activity. Interestingly, we found that the representations of target and salient distractor locations were not independent. Rather, they occupied a shared subspace suggesting a common substrate contributing to the formation of the priority map.

Behaviorally, the presence of a singleton distractor led to a decrease in the time required to locate the target. This behavioral benefit indicates that the salient stimulus was actively suppressed, effectively reducing the size of the search array [9]. A similar improvement or an absence of impairment in visual search in the presence of a singleton distractor has been extensively reported in the human literature and is mostly evident when subjects are in a "feature search mode" (i.e., they search for a particular feature in the presence of a singleton distractor) [1,9,34,35]. Thus, the behavior of our monkeys searching for a target whose features are known in advance is similar to that of humans. By contrast, when subjects are in a "singleton

detection mode" (i.e., searching for an oddball stimulus), the presence of a singleton distractor can significantly impair visual search (see [36] for a relevant monkey study; [1] for a review of the human literature). We suggest that the lack of an impairment and modest improvement in performance in the presence of the singleton distractor in our study is likely due to (i) prior experience with the task in which color singletons were irrelevant as well as (ii) top-down guidance offered by the cue indicating the target, at the beginning of each trial. Since the salient stimulus was never the target, the extended experience acquired during training and task execution possibly facilitated inhibition of the singleton distractor within the color dimension (i.e., the singleton color among uniformly colored items) [37]. Moreover, knowledge of the target features at the beginning of each trial could eliminate attention capture by the irrelevant singleton. Previous research has suggested that a color singleton can only be suppressed if its color is known in advance (i.e., first-order feature suppression [38]). However, since the singleton color in our task was not known beforehand, our results show that experience with the task can lead to the suppression of a singleton regardless of its color (i.e., second-order feature suppression [37,39,40]).

Previous FEF and LIP studies reported only singleton distractor suppression effects and no enhancement effect concluding that selection of salient stimuli by FEF and LIP is not automatic [15,16]. However, a pure salience response to a singleton stimulus has been described in LIP during a passive fixation task [41]. In our study, we identified distinct subpopulations, in both areas, that exhibited mixtures of temporal selectivity profiles, ranging from singleton suppression to early and sustained singleton enhancement. This suggests that both FEF and LIP contribute to the network encoding salient, albeit irrelevant stimuli (i.e., automatic selection) as well as to the suppression of such stimuli for goal-directed behavior. Although differences attributed to variations in tasks or subjects' behavior cannot be ruled out, our study shows the importance of assessing the heterogeneity of distinct population activity patterns in order to obtain mechanistic insights. Our results indicate that parallel mechanisms of salient distractor enhancement and suppression are implemented at the neuronal level in FEF and LIP. Interestingly, a brief early enhancement of the salient distractor followed by a later suppression has been recently reported in V4, in a task with an irrelevant color singleton [14]. The present study extends the findings of Kling and colleagues by showing that the early enhancement in activity in V4 for salient distractors is not unique to the visual cortex. It should also be noted that in a previous LIP study, enhanced responses to a single stimulus that signified an antisaccade to the opposite location were stronger compared to those to the same stimulus when it marked the goal of the saccade. This suggests that suppressing a saccade to a salient stimulus may also induce an enhanced response in LIP due to the increased cognitive load [42]. It is thus possible, although counterintuitive, that the enhanced response to the singleton distractor in our study, in subpopulations of FEF and LIP, contributes to the mechanism that facilitates the suppression of irrelevant locations. Unfortunately, given the limited number of neurons per cluster and the minimum number of trials required, there was insufficient statistical power to address whether the enhancement of activity for the salient distractor observed in specific subpopulations correlated with search time and target identification.

Nevertheless, we were able to compare the latency of the early singleton enhancement effects across FEF and LIP and found that these occurred at similar times in the 2 areas. This is in line with a previous study, which showed that in monkeys trained to detect a salient singleton, prefrontal and parietal neurons show significant enhancements in activity at similar latencies [43], contrary to earlier accounts positing that bottom-up attention signals arise earlier in the parietal cortex relative to the PFC [44]. By contrast, salient distractor suppression effects emerged earlier in the FEF, in line with its role in attentional selection [45–47]. Note that latency estimates vary according to the analysis approach. Here, we aimed to compare the

relative onset of salient distractor suppression and enhancement across FEF and LIP and thus, the reported latencies do not reflect the absolute latency of the effects in the 2 areas.

Our results are consistent with the signal suppression hypothesis, which postulates that salient stimuli generate a bottom-up "attend-to-me" signal that competes for attention, but this early salience signal can be overridden by an inhibitory process that prevents attentional capture. Event-related potential (ERP) studies have suggested that the bottom-up salience signal is reflected by an early positivity posterior contralateral (Ppc) component that appears before the canonical $P_D$ component, which indicates suppression see Gaspelin and colleagues in [2]. Thus, it is possible that the early enhancement and later suppression effects found in our subpopulations are linked to the pre-attentive and active suppression mechanisms postulated by the signal suppression hypothesis. An initial transient enhancement in activity followed by suppression could also be consistent with the rapid disengagement theory, which assumes that attention is initially captured by the salient stimulus followed by rapid disengagement of attention from the salient distractor location [4]. Assessing whether the early activity enhancement in our study reflects attentional capture or rather merely low-level sensory properties associated with the singleton would require additional task conditions and was beyond the scope of this study.

Our results suggest a sparser representation of information in LIP, as evidenced by the fact that we found a smaller proportion of units whose activity for the singleton distractor is suppressed. A previous neurophysiological LIP study demonstrated reduced responses at the population level to a salient compared to a non-salient distractor [16]. However, the salient stimulus in [16] served as a target during training sessions, and as a result it probably required greater effort to be ignored. It is possible that under conditions in which the salient stimulus has been rendered relevant and enhanced effort is required to filter it out, effects are more pronounced. Another difference between our task and that of previous studies [15,16], is that in our task the target identity was not known in advance and changed from trial to trial. It is possible that in conditions where the target is always the same, a feature search mode is stronger, resulting in an overall more prominent suppression of distractors.

Psychophysics and ERP studies have provided evidence for active suppression of singleton distractors relative to other distractors [9,35,48]; however, it has been debated whether such an effect is rather explained by an enhancement of target features shared by the non-singleton distractors [12,13]. This is a crucial question that pertains to the very idea of the signal suppression hypothesis. To address this issue, we exploited the fine spatial resolution afforded by invasive electrophysiological recordings, which allows manipulations of stimuli at the level of relatively small receptive fields, overcoming the difficulty associated with ERP studies where comparisons are carried out at the level of hemifields making manipulations of context while maintaining the stimuli identity harder. We directly compared the responses to a salient distractor in the RF to those with the exact same stimuli presented in a non-salient context. We found that in FEF, but not in LIP, singleton distractors are suppressed, relative to the same stimuli presented in a non-salient setting consistent with the signal suppression hypothesis. Thus, in addition to the well-documented role of the FEF in the enhancement of behaviorally relevant features [24–26,44], the FEF are also part of the circuit that actively suppresses salient but irrelevant distractors. The fact that this was not the case in LIP, suggests a less prominent role of the parietal area in the suppression of salient but irrelevant stimuli and a distinct contribution of FEF during visual search. This is in line with previous findings comparing the contribution of posterior parietal and dorsolateral prefrontal cortices in the representation and suppression of distractors [31,33]. Our results also contribute to a recent debate in the literature on whether signal suppression occurs only with singletons that are not salient enough to capture attention [49–51]. Our finding that a salient singleton is more strongly suppressed in

the FEF relative to the exact same stimulus presented in a non-salient context, provides additional support to the idea that the more salient the distractor, the stronger the suppression, in agreement with previous studies [50,51].

Although our results from activity averages point to differences in the way FEF and LIP contribute to the encoding and suppression of irrelevant salient stimuli, it is hard to infer from population averages how the target and salient distractor are represented at the population level. From a population-level perspective, cognitive functions have more recently been described by neural state spaces and the transition between different subspaces or the dynamics of activity within spaces (e.g., as captured by neural trajectories) [52]. In other words, different cognitive states arise as the system transitions to a different subspace [28] or as it moves to distinct locations in the state space. In this context, we suggest that both the suppression and facilitation effects that we report shape the population representations of the target and singleton distractor. However, even in this framework, it is important to assess how the activity of individual neurons with distinct selectivity properties (e.g., mixed selectivity) contributes to the population code [53].

To examine how salient distractors and targets are represented at the population level in FEF and LIP, we conducted decoding and dimensionality reduction analyses that provide insights about population-level representations. First, we found that the location of both target and salient distractor could be decoded from the pattern of population activity with high accuracy from either region. In addition, we observed that units with mixed selectivity (i.e., those encoding both target and singleton locations) had higher selectivity than those selective for either the target or the singleton distractor alone. This observation is consistent with recent FEF and PFC studies reporting higher spatial attentional modulation indices for neurons with mixed compared to classical selectivity [54,55]. Further, we found that mixed selective units drove to a large extent population-level decoding. These observations further highlight the prominence of mixed selectivity in neural representations [56].

In our paradigm, successful identification of the target location requires effective suppression of the singleton. A critical question is how the locations of the target and irrelevant salient distractor that can potentially capture attention are represented in the brain to avoid interference. One idea would be that the target and singleton distractor location are encoded by distinct patterns of population activity, resulting in orthogonal representations. Indeed, recent research has demonstrated that different memory items occupy independent subspaces, which could potentially provide a coding mechanism where competing items are represented independently, thus reducing interference [28]. Motivated by this finding, we asked whether the target and the salient distractor occupied distinct, low-dimensional subspaces. We observed that although independent target and singleton location representations could be identified, shared components that carried task relevant information also existed, indicating a lack of orthogonality. The shared representation suggests that the location representations of targets and irrelevant distractors are to some extent overlapping indicating a potential mechanism of interference from salient stimuli. It is possible, however, that representations were truly orthogonal when search was more efficient (e.g., in fast trials but less so in slow trials), something that we could not test in our data due to insufficient number of trials for the target and singleton at each location and condition (fast versus slow).

This lack of orthogonality may not be surprising, given the prominence of mixed selectivity in the 2 regions. However, we found that conjunctive units did not contribute larger weights to the shared subspace. This suggests that the shared subspace is composed by a weighted combination of neurons, including those with nonsignificant selectivity, rather than a distinct subpopulation. This aligns with other studies indicating that it is the pattern of activity across the population, rather than a segregated subpopulation, that identifies subspaces [29,30].

In this study, we investigated target and singleton distractor representations using activity average and population-level approaches. On the one hand, the traditional approach examines the level of excitation and inhibition that shapes the priority map and determines whether a distractor is effectively ignored or not. On the other hand, the more recent population-level approach examines the neural spaces and dynamics that determine which stimulus will be attended or ignored. In this framework, enhancement and suppression of activity are relevant to the extent that they shape the pattern of population activity. Reconciling these 2 approaches will be the next major challenge in neuroscience. Unfortunately, limitations in our study (e.g., the number of trials that were available for each unit) did not allow us to test several hypotheses that could have further illuminated aspects of this debate. For instance, it was not possible to examine the contribution of functional subpopulations in Fig 4 to the identified subspaces or to ascertain how the geometry of the subspaces was influenced by behavior during fast and slow trials. Our results do, however, provide novel insights into how behaviorally relevant and conspicuous, irrelevant—distracting—stimuli are represented, both at the global population level, as well as at the level of local neuronal activity in FEF and LIP, highlighting similarities and differences across the 2 areas. We show that both areas encode the irrelevant stimulus by a combination of activity enhancement and suppression across different subpopulations and that the representations of target and salient distractor are not orthogonal in either area. However, the FEF has a more prominent role in the suppression of the salient stimulus that leads to a more efficient identification of the target. We suggest that these variable activity patterns allow for rapidly adapting mechanisms that are implemented at the population level and underline the versatility of visual search depending on the current context.

## Materials and methods

### Subjects and surgical procedures

Two female rhesus monkeys (Macaca mulatta) weighing 4 to 6 kg were used in the study. Both animals were purposely bred by authorized suppliers within the European Union (Deutsches Primatenzentrum and Cyno Consulting). Experiments were carried out at facilities approved by the Veterinary Authorities of the Region of Crete (Medical School, University of Crete, EL91-BIOexp-06) and complied with the European (Directive 2010/63/EU and its amendments) and national (Presidential Decree 56/2013) laws on the protection of animals used for scientific purposes. Experimental protocols were approved by the Institutional Experimental Protocol Evaluation Committee (Approval 6170/7-5-2014).

The monkeys were implanted with a titanium post to fix the head. Following training, 2 titanium recording chambers were implanted, one over the FEF and one over LIP [stereotaxic coordinates: monkey PT: FEF, anteroposterior (AP) 27, mediolateral (ML) 17.5; LIP, AP −6, ML 12; monkey FN: FEF, AP 26, ML 18; LIP, AP −3, ML 12, based on MRI scans]. Surgical procedures were performed under general anesthesia and aseptic conditions.

### Behavioral tasks

Experimental and recording procedures were as described in [25]. Briefly, monkeys were seated in front of a CRT monitor (resolution, 800 × 600; refresh rate, 100 Hz) at a distance of 36 cm, inside a dark isolation box. Stimulus presentation and monitoring of behavioral parameters were controlled by the MonkeyLogic software package [57]. Eye position was monitored by an infrared-based tracking system (ETL-200; ISCAN) and was sampled at 120 Hz.

In salient distractor displays (Fig 1A), the stimulus popped out by virtue of its color, i.e., a red singleton in an array of green stimuli, a green singleton in an array of red stimuli, or a yellow singleton in an array of blue stimuli. All three different color configurations for target and

salient distractor were presented to both monkeys. The location and identity of the salient distractor were randomized in each trial and were not known beforehand. Salient distractor trials were randomly interleaved with displays in which the salient distractor was absent (i.e., all distractors shared the target's color, Fig 1B, or with mixed-color displays in which we kept the salient and the two flanking stimuli of Fig 1A constant and all other distractors shared no features with the target (Fig 1C). The shape of stimuli was pseudorandomly generated in each daily session from a pool of 8 shapes. Stimuli were $1.6 \times 1.6°$ in size and were matched for the number of pixels. Colors were matched for luminance ($\sim 7$ cd/m$^2$), except for the salient distractor that was made brighter in some sessions. Mixed-color trials were not run in sessions with a bright salient distractor. Nevertheless, results for the bright and non-bright salient distractors were similar and thus were combined. Stimuli were presented on a dark background (0.11 cd/m$^2$). To initiate the trial, monkeys had to fixate a square spot (0.75° × 0.75°) at the center of the screen for 300–500 ms within a 2.5° diameter window. Subsequently, the fixation spot was replaced by the cue, which indicated the target that the animals had to locate during the search. After 800 ms, the cue was replaced by the fixation spot and the monkeys were required to maintain central fixation for 500–1,000 ms. Following the delay period, a search array of 8 stimuli appeared. One of the stimuli was the target, and the remaining were distractors. Animals were allowed to freely move their eyes to scan the array to locate the target within 3.5 s. When they identified the target, they had to fixate it for 700 ms to receive water or juice reward. If monkeys broke fixation or failed to fixate the target within 3.5 s, the trial was aborted. Search array stimuli were presented in a rhombus arrangement at an eccentricity of 6–11° depending on the RF eccentricity of the recorded neurons. To map RFs, we used a memory guided saccade task as described elsewhere [25]. Briefly, in the memory guided saccade task, a target stimulus flashed for 150 ms in one of 24 possible locations. After a variable delay of 500–1,000 ms the fixation spot was switched off and animals were rewarded for making a saccade to the position of the target. The 24 possible positions were arranged in an inner (8 positions) and an outer (16 positions) rhombus. The inner rhombus comprised the 8 search array locations and included the preferred position for most recorded units. In the outer rhombus, stimuli were presented at eccentricities that were twice those of the inner rhombus.

## Recordings

Neural activity was simultaneously recorded from FEF and LIP using the Omniplex system (Plexon Inc). On a given day, up to 4 glass-coated tungsten microelectrodes (impedance, 1–1.5MΩ; Alpha-Omega Engineering) were positioned through a grid system over each area and were advanced through the dura by a four-channel microdrive system (NAN Instruments). Signals were filtered between 300 Hz and 8 kHz, amplified, and digitized at 40 kHz to obtain spike data. Spike waveforms were sorted off-line to isolate waveforms from single neurons using template matching and principal component analysis. If there was no clear isolation based on waveform projections on the principal component space or it was not possible to keep isolation on a single neuron throughout the entire session, multiunit activity was also accepted. In each session, electrodes were advanced through the dura until we recorded visual responses to flashing stimuli at 6°–12° eccentricities, in the fixation task. Subsequently, responses were recorded in the memory-guided saccade task and in the visual search task. The location of recordings in both FEF and LIP was estimated at the end of the experiments from MRI scans, the chambers' angles relative to the coronal and sagittal planes, and the depth of each recording site. FEF recordings were in the anterior bank of the arcuate sulcus, and LIP recordings in the lateral bank of the intraparietal sulcus. To ensure that parietal recordings did not include sites in 7a at the lip of the sulcus, LIP recordings were obtained at depths between

3 and 7 mm from the first activity encountered in each electrode, within the lateral bank of the intraparietal sulcus. To verify that our frontal recordings were in the FEF, we also electrically stimulated FEF sites at the end of the experiments in both monkeys and elicited eye movements with currents lower than 50 μA using 70-ms trains of biphasic pulses (duration, 500 μs) at 350 Hz [58].

## Data analysis

To determine saccade onset, we considered the horizontal (x) and vertical (y) eye position signals during a period ranging from the initial fixation to the offset of the array. We calculated the instantaneous $v_x$ and $v_y$ velocities and used a velocity threshold of 50°/s combined with an amplitude threshold of 0.4° for the detection of saccades.

**Firing rate and selectivity analysis.** Spike density functions were generated by convolving spikes with a Gaussian filter (sigma = 10 ms). To compare conditions at the population level, firing rates of individual units were normalized by dividing by the maximum response across conditions 40–200 ms following array onset. Only units with at least 10 trials were included in the analyses. All analyses were performed on the convolved spike data.

We defined visually responsive units as those with a significant visual response at the onset of the search array, relative to baseline (40–120 ms following array onset versus −150 to 0 ms before array onset, paired $t$ test, α = 0.05).

To obtain quantitative measurements of the effect size, a modulation index between 2 conditions was calculated for each unit (Figs 2B, 3B, and 5B) as follows: $(FR_{cnd1} - FR_{cnd2})/(FR_{cnd1} + FR_{cnd2})$, where FR is the average firing rate in the interval 150–200 ms following array onset. Effect sizes for Wilcoxon signed-rank and rank-sum tests were calculated as $r = z/\sqrt{n}$, where $n$ is the number of observations.

To assess the direction and magnitude of a neuron's firing rate difference between 2 conditions over time we utilized a receiver operating characteristic (ROC) analysis. We contrasted responses in the singleton versus matched stimuli in the RF in Fig 3, and fast versus slow trials in Fig 5. The ROC curve was estimated by calculating the proportion of trials in which the response to a condition was greater than a spike count criterion that varied from 0 to the maximum number of spikes in any time bin. After obtaining the ROC curve for the 2 conditions, the area under the curve (AUROC) was estimated for each unit (polyarea function in Matlab), reflecting the difference between conditions. A null AUROC was generated for each unit by shuffling trial indices between conditions (500 shuffles). To compare the average AUROC with chance, we generated a null distribution for the population by randomly selecting a null value from each unit and then calculating a null average across units. We repeated this procedure 1,000 times, producing a distribution of 1,000 null values for each time point. We compared this with the observed AUROC and generated time-resolved $p$-values. To correct for multiple comparisons across time, we applied the procedure described in [59] for false discovery rate.

To quantify the information carried in the responses of individual units for target and singleton location, we used the PEV statistic [60]. PEV reflects the amount of variance in the unit's firing rate that can be explained by the different task variables. In accordance with previous studies [61,62], we used the $\omega^2$ statistic that is an unbiased measure of explained variance defined as follows: $\omega^2 = (SS_{between\ groups} - df \times MSE)/(SS_{total} + MSE)$, where df is degrees of freedom, $SS_{between\ groups}$ is the sum of squares between groups (levels), $SS_{total}$ is the total sum of squares, and MSE is the mean square error. These quantities were calculated for each unit by performing a one-way ANOVA across trials with factors either the target or the salient distractor location (5 locations). To quantify information in a time-resolved manner, spike counts

were calculated within 50-ms sliding windows, advanced in 10-ms steps, and $\omega^2$ was computed in each window. For each unit, a null distribution of $\omega^2$ values was estimated by shuffling the trials corresponding to different locations and repeating this procedure 200 times. Subsequently, PEV values were z-scored by subtracting the mean and dividing by the SD of the null distribution. We determined the significance of individual units based on whether the PEV exceeded 1.645 (one-tailed test) in the interval 150–200 ms following array onset.

**Clustering of temporal selectivity profiles.** We identified neuronal subpopulations based on the structure of their temporal selectivity profiles, as quantified by a t-value that reflects a unit's selectivity to a salient versus a non-salient distractor within the RF. Then, we applied the PhenoGraph clustering algorithm [27], which is an efficient graph-based method for identifying neuronal subpopulations with similar properties in high-dimensional data. The method has recently been employed to sort neuronal selectivity profiles for sequences of sounds in the mouse auditory cortex [63]. Briefly, the algorithm takes as input an N × D matrix where N are the neurons and D the t-values over time. First, the algorithm identifies for each neuron its K nearest neighbors (based on the similarity of temporal selectivity profiles) using Euclidean distance, where K is the only input parameter to the method. This procedure defines N sets (one for each neuron) in which each cell is connected to K neighbors. Second, a graph is built in which the weight of the edge between every pair of nodes (neurons) is based on the number of neighbors they share. In our case, neurons with similar temporal selectivity profiles constitute a community that is characterized by the density of their interconnections, i.e., its modularity. Subsequently, the algorithm partitions the graph into communities that maximize modularity. The main advantage of PhenoGraph clustering over other methods is that it does not require a priori specification of the number of clusters and is especially powerful for high dimensional data. While we used K = 40 as input to the algorithm, different values of K produced similar results. We considered units with significant visual responses (paired *t* test, $p < 0.05$, −150–0 ms before array onset versus 40–120 ms following the array onset) and excluded those with noisy responses as reflected by differences between the 2 conditions during the baseline (*t* test, $\alpha = 0.1$, −150–0 ms before array onset).

**Decoding analysis.** A linear SVM classifier was used to decode task-relevant information from neuronal responses using the LIBSVM library (version 3.17) [64]. Estimates of decoding performance were based on neuronal populations obtained by combining units recorded over different sessions. Although this approach does not take into account the correlated activity between neurons, the main findings of this study still hold given that we are primarily interested in comparisons between brain regions and stimulus conditions. To decode whether the target or a non-salient distractor was in the RF, or whether the salient or a non-salient distractor was in the RF (Fig 2C), we considered trials in which the first saccade was made away from the RF. Chance accuracy in this case was 50%. For the decoding of target and salient distractor location, we considered positions contralateral to the recording hemisphere, i.e., the 5 left hemifield locations (including the 2 vertical positions), thus chance accuracy was 20%. The same approach was followed for selecting data in the subspace analysis. Classification accuracy was estimated using a 5-fold cross-validation procedure [65]. To ensure equal prior probabilities, we stratified the number of trials by randomly selecting 10 trials in each condition by means of a resampling procedure (50 resamples). Before classification, data were scaled between 0 and 1 to ensure that classification was not influenced by the absolute magnitude of the responses. Train and test sets were not scaled independently but the same scaling was applied to both. To compute the time course of decoding accuracy, we used spike counts within 50 ms windows advanced in 10 ms steps. To ensure that differences in the decoding performance between the 2 areas were not due to differences in the size of the 2 populations, we stratified the number of units in the 2 areas using a resampling procedure (50 resamples).

Yet, using the full populations resulted in qualitatively and statistically similar results. The input to the classifier in each time window was an N × T matrix, where N is the number of units and T is the number of samples (trials × conditions). Typically, the number of units, N, was higher than the number of samples, T. The fact that N is greater than T can be detrimental for many classifiers due to the "curse of dimensionality" [65,66]. However, kernel methods such as SVM are more resilient to this problem as is evident from the high decoding accuracies obtained in this study.

Significant differences in decoding performance between conditions were assessed using a two-sided cluster-based permutation test [67] to correct for multiple comparisons over time (a = 0.05, $10^3$ permutations). To compare decoding accuracies with chance, we computed a null accuracy distribution for each time window by shuffling the trial indices. For each time window, the shuffling was repeated 500 times. We then derived $p$-values across time by comparing the actual to the shuffled performance. To correct for multiple comparisons over time, we used a cluster-based test applied to these $p$-values ($\alpha = 0.05$). For non-time resolved decoding, we estimated accuracy in the 150–200 ms window after array onset. We resampled trials and units using 100 resamples and performed shuffling 1,000 times. Statistical comparisons were carried out using permutation tests.

**Subspace analysis and optimization.** Subspaces were computed in the 150–200 ms interval following the array onset. The input to the subspace analysis was a 255×N neuronal population matrix, where the first dimension corresponded to the number of locations (i.e., 5) times the number of time points (i.e., 51), with N being the number of neurons.

We conducted 3 subspace optimizations using the method and code provided by [68]. The manopt MATLAB toolbox was employed to implement these optimizations [69].

We defined the **orthogonal subspaces** ($Q_{Orth-Target}$ and $Q_{Orth-Singleton}$) by ensuring that they were mutually orthogonal while maximizing the target and salient distractor variance, respectively. This was achieved by solving the following optimization problem:

$$\underset{Q \in M_{d+d}(\mathbb{R}^N)}{\text{maximize}} \frac{Tr(Q_{Orth-Target}^T C_{Target} Q_{Orth-Target})}{\sum_{i=1}^{d} \lambda_i^{Target}} + \frac{Tr(Q_{Orth-Singleton}^T C_{Singleton} Q_{Orth-Singleton})}{\sum_{i=1}^{d} \lambda_i^{Singleton}}$$

where $Q_{Orth-Target} \in \mathbb{R}^{N \times d}$ and $Q_{Orth-Salient} \in \mathbb{R}^{N \times d}$ are the Orth-Target and Orth-Singleton subspaces, respectively (i.e., d-dimensional bases over N neurons), $C_{Target}$, $C_{Singleton}$ are N×N covariance matrices that correspond to target and salient distractor activity, $\lambda_i^{Target}$, $\lambda_i^{Singleton}$ the sorted eigenvalues of $C_{Target}$ and $C_{Singleton}$ and $Tr$ is the matrix trace. This is an optimization over the Stiefel manifold.

The **exclusive subspaces** ($Q_{Excl-Target}$ and $Q_{Excl-Singleton}$) were generated by maximizing the target's (singleton's) variance, while keeping the singleton's (target's) variance bellow a small number $v$. We opted for $v = 0.01$ (i.e., 1%) as in [68]. Thus, exclusive subspaces were found by solving the following optimization problem:

$$\underset{Q_{Excl-Target} \in M_d(\mathbb{R}^N)}{\text{maximize}} \frac{Tr(Q_{Excl-Target}^T C_{Target} Q_{Excl-Target})}{\sum_{i=1}^{d} \lambda_i^{Target}},$$

$$subject\ to\ \frac{Tr(Q_{Excl-Singleton}^T C_{Singleton} Q_{Excl-Singleton})}{\sum_{i=1}^{d} \lambda_i^{Singleton}} \leq v$$

The **shared subspace** ($Q_{Shared}$) was then identified as being orthogonal to the exclusive subspaces, while jointly maximizing the variance of both target and singleton. In other words, the shared subspace solely comprised shared representations, while excluding target or singleton

exclusive ones. This was achieved by solving the following optimization:

$$\underset{Q_{shared} \in M_d(\mathbb{R}^N)}{\text{maximize}} \frac{Tr(Q_{Shared}^T C_{Target} Q_{Shared})}{\sum_{i=1}^{d} \lambda_i^{Target}} + \frac{Tr(Q_{Shared}^T C_{Singleton} Q_{Shared})}{\sum_{i=1}^{d} \lambda_i^{Singleton}},$$

$$\text{subject to } Q_{Shared} \perp Q_{Excl-Target} \text{ and } Q_{Shared} \perp Q_{Excl-Singleton}$$

We calculated the stimulus variance of projected activity captured by a subspace using the alignment index [70], a normalized variance metric that was defined as follows:

$$A = \frac{Tr(Q^T C Q)}{\sum_{i=1}^{d} \lambda_i}$$

where $Q$ is the subspace and $C$ the covariance matrix. Since $\sum_{i=1}^{d} \lambda_i$ is the maximum variance that can be captured in $d$ dimensions ($d = 5$ in our case), the alignment index takes values between 0 and 1. Unit contributions to a subspace were estimated by computing the magnitude of the weights, which is equal to the norm of each row of subspace $Q$. This is equivalent to the square root of the variance of the unit's activity projected onto the subspace, divided by the total variance of the unit [68].

## Supporting information

**S1 Fig. Average population responses aligned to the onset of the first saccade.** Responses were calculated in singleton present displays (example shown on the left). Normalized firing rates in singleton present displays with the singleton (green) and non-singleton (magenta) distractor in the RF, in FEF (left) and LIP (right). Error bars (shaded area around each line) represent ±SEM. The horizontal line at the top of each graph indicates periods with significant differences between the 2 conditions (permutation test, $p < 0.05$). Source data are available at https://zenodo.org/records/14577123.
(TIF)

**S2 Fig. Effect of singleton distractor on neuronal firing shown for each animal separately.** Responses in singleton and non-singleton distractor in RF were contrasted in singleton present displays (example display shown on top left). (A) Data for monkey PT. Top panels show normalized population average firing rates aligned to array onset in singleton present displays, with the singleton (green) and a non-singleton (magenta) distractor in the RF, in FEF (left) and LIP (right). Error bars (shaded area around each line) represent ±SEM. The horizontal line at the top of each graph indicates periods with significant differences between the 2 conditions (permutation test, $p < 0.05$). Bottom panels show distribution of modulation indices quantifying the difference between responses to the singleton and non-singleton distractors (150–200 ms following array onset) at the level of individual units in FEF (left) and LIP (right). Colored bars correspond to units with significant suppression (green) or enhancement (magenta) of responses for the singleton distractor (two-sample $t$ test, $p < 0.05$). Arrows indicate the median of each distribution (FEF: −0.05; $p < 0.001$, LIP: −0.035, $p < 0.01$; Wilcoxon rank-sum test). (B) Data for monkey FN. Same conventions as in (A). The medians of each distribution are FEF: −0.03; $p < 0.01$, LIP: 0.005, $p = 0.8$; Wilcoxon rank-sum test. Source data are available at https://zenodo.org/records/14577123.
(TIF)

**S3 Fig. Distinct subpopulations with unique selectivity profiles shown for each monkey separately.** Responses in singleton present and singleton absent displays (example displays

shown on top left) were contrasted. (A) The 3 clusters identified by the PhenoGraph algorithm for (A) monkey PT and (B) monkey FN. Graphs show average normalized firing rates in singleton (green) and non-singleton (magenta) in the RF trials. Error bars (shaded area around each line) represent ±SEM. Source data are available at https://zenodo.org/records/14577123. (TIF)

**S4 Fig. Distinct subpopulations with unique selectivity profiles.** Responses were calculated from singleton-present displays (example shown on the bottom right). (A) The PhenoGraph algorithm identified 4 FEF subpopulations. Two of these subpopulations exhibited singleton suppression (clusters 1 and 3), one subpopulation showed singleton enhancement (cluster 4), and one exhibited an early enhancement followed by a later suppression (cluster 2). Panels show average firing rates in singleton (green) and non-singleton in RF (magenta) trials. Horizontal line at the top indicates periods with significant differences between the 2 conditions (permutation test, $p < 0.05$). (B) The clustering algorithm identified 3 LIP subpopulations. One exhibited singleton suppression (cluster 1), a second one showed singleton enhancement (cluster 2), and a third one an early enhancement followed by a subsequent suppression (cluster 3). Source data are available at https://zenodo.org/records/14577123. (TIF)

**S5 Fig. LIP activity following array onset in fast and slow trials.** Responses were calculated in singleton present displays (example shown on the top left). (A) LIP responses to singleton (left) and non-singleton (right) distractor in RF in fast (solid line) and slow trials (dashed line). (B) Modulation indices quantifying the difference between fast and slow trials at the level of individual LIP units for singleton (left) and non-singleton (right) distractors in the RF. Arrows indicate the median of each distribution. Source data are available at https://zenodo.org/records/14577123. (TIF)

**S6 Fig. Contribution of mixed selective neurons to encoding of target/singleton location.** Responses were calculated in singleton present displays (example shown on top left). (A) Decoding of target location from FEF (left) and LIP (right) activity in the 150–200 ms interval after array onset. Violin plots show the distribution of decoding accuracies calculated over 50 resamples across different trials. The dashed horizontal line indicates chance accuracy. The 3 left hemifield and 2 vertical meridian locations were considered, so chance was at 20%. Crosses indicate the mean of each distribution. Blue stars above each distribution indicate significant accuracy relative to chance (permutation test). Stars above horizontal bars indicate difference between areas (permutation test). (B) Same for singleton location. Source data are available at https://zenodo.org/records/14577123. (TIF)

**S7 Fig. Contribution to the shared subspace of LIP units with different selectivity.** Violin plots show the distribution of weights contributing to the shared subspace for each category. Crosses represent the mean of each distribution and boxes the median. Data were obtained from the singleton present displays (shown on top right). Source data and relevant code are available at https://zenodo.org/records/14577123. (TIF)

## Author Contributions

**Conceptualization:** Panagiotis Sapountzis, Georgia G. Gregoriou.

**Data curation:** Panagiotis Sapountzis.

**Formal analysis:** Panagiotis Sapountzis, Alexandra Antoniadou.

**Funding acquisition:** Panagiotis Sapountzis, Georgia G. Gregoriou.

**Investigation:** Panagiotis Sapountzis, Georgia G. Gregoriou.

**Methodology:** Panagiotis Sapountzis, Georgia G. Gregoriou.

**Project administration:** Panagiotis Sapountzis, Georgia G. Gregoriou.

**Resources:** Panagiotis Sapountzis.

**Software:** Panagiotis Sapountzis, Alexandra Antoniadou.

**Supervision:** Georgia G. Gregoriou.

**Validation:** Panagiotis Sapountzis.

**Visualization:** Panagiotis Sapountzis.

**Writing – original draft:** Panagiotis Sapountzis, Georgia G. Gregoriou.

**Writing – review & editing:** Panagiotis Sapountzis, Alexandra Antoniadou, Georgia G. Gregoriou.

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
