## [Editor Report · Decision Letter 0]

29 Jul 2024

Dear Dr Gregoriou, 

Thank you for submitting your manuscript entitled "Diverse activity profiles contribute to the control of distraction in the prefrontal and parietal cortex" for consideration as a Research Article by PLOS Biology.

Your manuscript has now been evaluated by the PLOS Biology editorial staff and I am writing to let you know that we would like to send your submission out for external peer review. 

Please note that we unfortunately have not been able to receive advice from one of our academic editors on your study and have, therefore, not yet made a firm decision on whether the conceptual advance is sufficient for PLOS Biology. We will discuss this after review with one of our editorial board members and will be looking for strong reviewer support.

Once your full submission is complete, your paper will undergo a series of checks in preparation for peer review. After your manuscript has passed the checks it will be sent out for review. To provide the metadata for your submission, please Login to Editorial Manager (https://www.editorialmanager.com/pbiology) within two working days, i.e. by Jul 31 2024 11:59PM.

Kind regards,

Christian

Christian Schnell, PhD

Senior Editor

PLOS Biology

cschnell@plos.org

---

## [Decision Letter · Decision Letter 1]

10 Oct 2024

Dear Georgia,

Thank you for your patience while your manuscript "Diverse activity profiles contribute to the control of distraction in the prefrontal and parietal cortex" was peer-reviewed at PLOS Biology. Apologies for the long delay in getting back to you, it was a bit more challenging than usual to find suitable reviewers and I was attending conferences in the last two weeks, so am a bit behind with 'my' manuscripts. In any case, your manuscript has now been evaluated by the PLOS Biology editors, an Academic Editor with relevant expertise, and by several independent reviewers. 

In light of the reviews, which you will find at the end of this email, we would like to invite you to revise the work to thoroughly address the reviewers' reports.

As you will see below, the reviewers find your study interesting and that it provides important insights. They mostly ask for a few additional analyses and clarifications.

Given the extent of revision needed, we cannot make a decision about publication until we have seen the revised manuscript and your response to the reviewers' comments. Your revised manuscript is likely to be sent for further evaluation by all or a subset of the reviewers.

**IMPORTANT - SUBMITTING YOUR REVISION**

*Re-submission Checklist*

*Published Peer Review*

*PLOS Data Policy*

*Blot and Gel Data Policy*

Sincerely,

Christian

Christian Schnell, PhD

Senior Editor

PLOS Biology

cschnell@plos.org

REVIEWS:

Reviewer #1: This is an interesting article examining how neurons in the frontal and parietal lobe represent task relevant and irrelevant stimuli. The frontal eye fields (FEF) and the lateral intraparietal area (LIP) have been thought to represent salient stimuli and filter distractors, with similar magnitude and time course. Recent results in the visual cortex, and technical developments in the analysis of neural population data have prompted the authors of this study to reconsider the question. They therefore have trained monkeys to perform a very difficult behavioral task which involves the selection of stimuli based on shape and color. Their results reveal a second population of neurons, whose activity is enhanced by the distractor, and systematic differences between the two areas. Furthermore, they show that representation of targets and salient distractors occupy a shared subspace. The results are novel and interesting and make a valuable addition to this literature. The manuscript is written clearly and is accessible to a wide audience, despite the highly technical nature of some results and analyses. There are some issues however that need to be addressed before the manuscript can be published. 

1. The authors begin the presentation of their results with the comparison of the distractor suppression between areas. This result is probably the weakest of all the effects presented in the paper. Both FEF and LIP exhibit a pattern of singleton distractor suppression that is qualitatively similar. The magnitude of the FEF population suppression in figure 2B-left seems very small and the distribution of modulation indexes between the two areas in figure 2B-left and -right almost identical. Assuming that the reported statistics are correct as reported, the authors should include effect sizes and provide some intuition of how the comparison between the two samples that look so similar reached significance. In any case, toning down the presentation of this result would probably help the paper: in both areas there are neurons for which activity of the singleton distractor is suppressed - this is what can be unequivocally gleaned from this analysis. 

2. The latency of the effect could have been better explored. As the authors are aware, it has been proposed that signals of bottom-up attention appear earlier in LIP than FEF, whereas top-down attention follows the reverse sequence (an idea championed by reference 43). Compelling evidence has been presented since that the former is not true. The authors have the opportunity to disprove the latter claim as well, as their paradigm is a prototypical top-down attention task that requires subject to actively ignore bottom-up signals. 

3. There is no discussion of results from the two individual monkeys in either the first or subsequent result sections. Some confirmation of the main findings of the study across both individuals would strengthen the paper.

4. Page 6, last sentence of the last complete paragraph: this point was not clear. Was the salient distractor suppression not evident or decodable for FEF? 

[Revision along the lines suggested in point 1 may make this comment moot].

5. Describing a color as "petrol" may not be clear for all readers. 

6. The authors include units in the selectivity analysis of page 9 if they exhibited significant t-values for 10 consecutive 1-ms bins. The choice of bin duration seems very short. How many trials were available to provide such a level of precision? 

7. Page 11, first sentence. The authors do not quite show such a strong, inverse relationship. This statement should be rephrased. 

8. Findings of mixed selectivity were very interesting. In recent years, a lot of attention has been devoted particularly to nonlinear mixed selectivity i.e. selectivity for a stimulus dimension that changes nonlinearly across different contexts. It would be interesting for the authors to further break down their results in units that show linear and nonlinear mixed selectivity. 

9. Page 14, first paragraph of Discussion section: In discussing the roles of prefrontal and posterior parietal cortex more broadly, the authors may wish to refer to Qi et al. (2015), J Neurophys 113(1):44-57. 

Reviewer #2 (Xiaojin Ma): This is an interesting study using single cell recording to pinpoint the neural activities in the FEF and LIP regions during singleton distractor suppression. As a disclaimer, my research involves electrophysiological techniques but unfortunately not single cell recording. Therefore, my comments are mainly on the appropriateness of experiment design to study attentional suppression, but not on the technical aspects unique to single cell recording or analysis.

The choice of the FEE and LIP do seem to be relevant to the study of attentional suppression. However, in the Introduction I hoped to see more explicit a priori hypotheses about how the activity patterns of the two regions would correspond to implications on different attributes of the mechanism of attentional suppression. This would pave the way for later interpreting differential neural responses in the two regions.

The experiment had the target identity change from trial to trial, which may have a few undesired effects. First, the frequent change in the top-down goal could possibly weaken the feature search mode that is necessary for suppression to occur. Second, the target on the 1-back trial may be a non-singleton distractor on the current trial. Intertrial priming of the target and the general selection history of every non-singleton shapes could inflate the target feature enhancement effect that was measured by saccades to the non-singleton. This may to an extent explain why the present study were able to find both suppression and enhancement effects in the two brain regions whereas the previous ones that used a fixed target identity only found suppression. I wonder what the consideration is behind the authors' decision to vary the target shape.

As the authors mentioned in the paper, FEF and LIP are known for encoding saliency. It is thus important to distinguish in the results whether the decoded location information about the singleton reflect detection of its saliency or subsequent suppression of it utilizing its saliency signal. The latter was to an extent supported by the reduced firing rate to a singleton compared to a target in neural activity. The decoding results may also contain similar information, and it would be great to emphasize such distinction when interpreting the results. ERP studies have shown that the component suggesting suppression (PD) actually appears highly similar to the component suggesting saliency detection (Ppc), see a recent review paper on the PD component: Gaspelin et al (2023, J Cog Neuro).

In the Method section, there mentioned 3 different target-distractor color combinations, but only two monkey served as subjects. I couldn't find information about how the colors were assigned/counterbalanced and whether the assignment remained constant throughout the task for each monkey. Please specify these important information. The Vatterott, Mozer, & Vercera (2018, AP&P) and Gaspelin & Luck (2018, JEPHPP) papers showed that fixed color assignments to both the target and the singleton are critical for suppression to consistently occur.

References:

Gaspelin, N., Lamy, D., Egeth, H. E., Liesefeld, H. R., Kerzel, D., Mandal, A., ... & van Moorselaar, D. (2023). The distractor positivity component and the inhibition of distracting stimuli. Journal of cognitive neuroscience, 35(11), 1693-1715.

Vatterott, D. B., Mozer, M. C., & Vecera, S. P. (2018). Rejecting salient distractors: Generalization from experience. Attention, Perception, & Psychophysics, 80, 485-499.

Gaspelin, N., & Luck, S. J. (2018). Distinguishing among potential mechanisms of singleton suppression. Journal of Experimental Psychology: Human Perception and Performance, 44(4), 626.

Reviewer #3: Review of Sapountzis, Antoniadou, & Gregoriou, "Diverse activity profiles contribute to the control of distraction in the prefrontal and parietal cortex"

Manuscript ID: PBIOLOGY-D-24-02168R1

Summary

 The authors tested whether salient color singleton distractors (e.g., a red item among all green items) are suppressed by the early visual system using recordings of rhesus monkeys in areas FEF and LIP. To test suppression, the authors presented the monkeys with singleton present trials and singleton absent trials with the task of fixating a specific-feature (e.g., a diamond shape) target among heterogeneously shaped distractors (e.g., stars, circles, etc.). Neuronal populations in FEF and LIP were recorded during the task and analyzed in many ways to support the overall notion that the color singleton distractors were suppressed, and this suppression cannot be explained by target feature enhancement. Further, the authors present evidence of different population types and areas of representation while also demonstrating the seemingly different roles of FEF and LIP in attentional selection.

 Overall, the manuscript was a real pleasure to read! It was well written, easy to follow, and tests a fundamental and important question concerning distractor suppression and the neurobiology associated with it. The authors had clear predictions and logic. To the authors' credit, there were many claims tested and despite some of the complexities, the results were straightforward and surprisingly easy to interpret. I have a few critiques listed below, which to be clear, are meant to be (mostly) read as suggestions (with the aim of strengthening the connections to the literature), not precursors preventing publication. 

Critique (in no particular order)

1) The authors test between target enhancement and distractor suppression, which was a welcome addition and a really cool finding (especially given the results suggest suppression). What might be potentially interesting would be to also relate these findings to recent claims about visual salience (e.g., Wang & Theeuwes, 2020 and the replies from Stilwell et al. 2023 & 2024). Briefly, the claim from stimulus-driven accounts was that the singletons were not salient enough to warrant suppression. Stilwell and colleagues developed methods to assess salience and in doing so, found evidence that the more salient the distractors became the greater the suppression (i.e., oculomotor suppression and probe-suppression). In the current study, it appears this may also be the case and would really complement recent support of the signal suppression hypothesis. The mixed color displays should render the singleton as less salient given less color contrast between the singleton and other items compared with the non-mixed displays. It appears that in Figure 3 the results support the claim that the more salient singleton (non-mixed displays) was suppressed more strongly than the mixed-color displays. Perhaps the authors could mention this finding and how it relates to signal suppression, especially recent claims of the role of salience. 

2) In a similar vein to the previous comment, if you look at Figure 3C, for example, there also seems to be evidence that FEF might track the earlier saliency computations. Recently, there has been evidence that salient stimuli produce an EEG/ERP, Ppc component which has been argued to be a "saliency detector" (John McDonald and colleagues as well as Hermann Muller's group have been pushing this idea recently). In unpublished data from my own research, we manipulated the salience of the singleton distractor via color contrast and found a modulation of this early positivity (likely the Ppc) both in amplitude (~0.5 uV) and in timing (~100 ms post display onset) as a function of salience, prior to the canonical PD component indicative of suppression. The Ppc was both larger and earlier for the high- than low-salience singleton, yet the following PD component was the same. In the current study, especially in Figure 3C, it appears the FEF shows a reliable "blip" at approximately 100 ms for the AUROC analysis, followed by suppression. There also seem to be hints of this early (~100 ms) pattern in the firing rates of FEF as well (e.g., in Figure 3A the singleton distractor has a higher rate of firing at about 100 ms than the matched stimuli). This might support the idea that the early salience signals can be modulated (and reflected in changes to FEF populations), but the subsequent suppression is stronger for these more salient distractors. It seems like the responsive unit analyses (i.e., those depicted in Figure 4) seem to also support these notions (greater representation of the salient singleton early on). These might be fruitful analyses/discussion points to further cement the current findings in support of the signal suppression hypothesis and more recent findings regarding the role of salience, namely that there is pre-attentive salience processing (what used to be referred to as the "attend-to-me" signal) followed by active suppression (often termed "proactive" suppression, though it's better stated as prior to the first shift of attention).

3) However, one thing to consider with the early enhancement followed by suppression results is the case of "rapid disengagement" which is argued by stimulus-driven accounts to explain all the suppression data (e.g., Theeuwes, 2010; 2024). The initial salience processing is selective in nature, and all suppression after this is "reactive" in nature. The clustering analyses seem to challenge if not refute the "early selection followed by rapid disengagement" idea given that some clusters show early suppression effects which seem plausibly too early for an initial shift of attention followed by rapid disengagement. Perhaps the authors could acknowledge this alternative and how the current study rules this out. 

4) The task used multiple singleton distractor colors which may have promoted "second-order" suppression (e.g., Gaspelin & Luck, 2018; Ma & Abrams, 2023; Won, Kosoyan, & Geng, 2019) in which case all of the salient singleton distractors were suppressed regardless of their specific colors. In the most recent version of the signal suppression hypothesis, the proponents argue the suppression is feature-specific (i.e., first-order in nature, e.g., Gaspelin & Luck, 2018, JEP:HPP). Perhaps the authors could briefly mention that the current patterns of suppression may be evidence for second- instead of first-order suppression. 

Minor points

1) Many of the figures with lines are difficult to see differences given the thickness of the lines. For example, in the figures depicting normalized firing rates (most Panel A's in the figures) it's difficult to discern (at least visually) where the conditions overlap and where they diverge. Also, the dashed lines have a lot of space between the dashes making it difficult to see the shape of the curve, perhaps the authors can decrease the dash spacing. 

2) Perhaps I missed this but why did the authors choose to analyze the time window from 150-200 ms, specifically? Is there an a priori reason to choose this window? For example, many (especially those in support of the signal suppression hypothesis using EEG/ERP) choose the start of the time window for the "early" PD component closer to 100 ms (~115-125 ms).

---

## [Editor Report · Decision Letter 2]

13 Dec 2024

Dear Dr Gregoriou,

Thank you for your patience while we considered your revised manuscript "Diverse activity profiles contribute to the control of distraction in the prefrontal and parietal cortex" for publication as a Research Article at PLOS Biology. This revised version of your manuscript has been evaluated by the PLOS Biology editors, the Academic Editor.

Based on our Academic Editor's assessment of your revision, we are likely to accept this manuscript for publication, provided you satisfactorily address the following data and other policy-related requests:

* We would like to suggest a different title to improve its accessibility for our broad audience: Diverse neuronal representation levels contribute to the control of visual distraction in the prefrontal and parietal cortex

* Please add the links to the funding agencies in the Financial Disclosure statement in the manuscript details.

* Please note that per journal policy, the model system/species studied should be clearly stated in the abstract of your manuscript. 

* DATA POLICY:

Regardless of the method selected, please ensure that you provide the individual numerical values that underlie the summary data displayed in the following figure panels as they are essential for readers to assess your analysis and to reproduce it: 2B, 3B, 5B, 6A, 7ABF, S2AB, S5B, S6 and S7.

* CODE POLICY

We expect to receive your revised manuscript within two weeks. 

*Published Peer Review History*

*Press*

Sincerely,

Christian

Christian Schnell, PhD, 

Senior Editor

cschnell@plos.org

PLOS Biology

---

## [Editor Report · Decision Letter 3]

8 Jan 2025

Dear Georgia,

Happy New Year!

Thank you for the submission of your revised Research Article "Diverse neuronal activity patterns contribute to the control of distraction in the prefrontal and parietal cortex" for publication in PLOS Biology. On behalf of my colleagues and the Academic Editor, Christopher Pack, I am pleased to say that we can in principle accept your manuscript for publication, provided you address any remaining formatting and reporting issues. These will be detailed in an email you should receive within 2-3 business days from our colleagues in the journal operations team; no action is required from you until then. Please note that we will not be able to formally accept your manuscript and schedule it for publication until you have completed any requested changes.

While you attend to those requests, please also either upload the S1 file or change the references in the figure legends to the zenodo repositories. Currently, you write that the source data can be found in the S1_data, but the the file has not been provided.

PRESS

Sincerely, 

Christian

Christian Schnell, PhD

Senior Editor

PLOS Biology

cschnell@plos.org